# Undiscovered bird extinctions obscure the true magnitude of human-driven extinction waves

Rob Cooke [1,2,3] ✉, Ferran Sayol[2,3,4], Tobias Andermann [5], Tim M. Blackburn [6,7], Manuel J. Steinbauer [8], Alexandre Antonelli [2,3,9,10] & Søren Faurby [2,3]

Birds are among the best-studied animal groups, but their prehistoric diversity is poorly known due to low fossilization potential. Hence, while many human-driven bird extinctions (i.e., extinctions caused directly by human activities such as hunting, as well as indirectly through human-associated impacts such as land use change, fire, and the introduction of invasive species) have been recorded, the true number is likely much larger. Here, by combining recorded extinctions with model estimates based on the completeness of the fossil record, we suggest that at least ~1300–1500 bird species (~12% of the total) have gone extinct since the Late Pleistocene, with 55% of these extinctions undiscovered (not yet discovered or left no trace). We estimate that the Pacific accounts for 61% of total bird extinctions. Bird extinction rate varied through time with an intense episode ~1300 CE, which likely represents the largest human-driven vertebrate extinction wave ever, and a rate 80 (60–95) times the background extinction rate. Thus, humans have already driven more than one in nine bird species to extinction, with likely severe, and potentially irreversible, ecological and evolutionary consequences.

Humanity's dispersal out of Africa and subsequent peopling of essentially all ice-free land across the globe has triggered waves of extinctions, which for many groups, including birds, have been particularly large across isolated archipelagos[1–3] (Fig. 1). Even small human populations rapidly devastated island avifaunas as they introduced new threats outside the evolutionary experience of native species[2]. Drivers of human-driven bird extinctions include habitat loss associated with land clearance (cutting, burning) and the introduction of non-native plants and crops[4], the introduction of alien species (domestic animals and/or human commensals)[2,5,6] and the

overexploitation of bird species via hunting and trapping (birds were hunted for their fat, protein, bones and colourful feathers)[1,5]. Yet, due to the incomplete avian fossil record[3,5,7,8], many bird extinctions are likely to have gone undiscovered—i.e., not yet found due to a lack of research effort or because they left no trace in the discoverable fossil record[4,7,9–12]. Hence, the full extent of bird extinctions since the Late Pleistocene remains unknown (although see previous studies of the Pacific[4,7,10,12]). Previous analyses of bird extinction rate have therefore focused on well-recorded observed extinctions[2,13,14], i.e., since 1500 CE (Common Era). However, ignoring fossil and undiscovered extinctions

[1]UK Centre for Ecology & Hydrology, Maclean Building, Crowmarsh Gifford, Wallingford, Oxfordshire OX10 8BB, UK. [2]Department of Biological and Environmental Sciences, University of Gothenburg, Box 463, SE-405 30 Göteborg, Sweden. [3]Gothenburg Global Biodiversity Centre, Box 461, SE-405 30 Göteborg, Sweden. [4]CREAF, E08193 Bellaterra (Cerdanyola del Vallès), Catalonia, Spain. [5]Department of Organismal Biology, SciLifeLab, Uppsala University, Uppsala, Sweden. [6]Centre for Biodiversity and Environment Research, Department of Genetics, Evolution and Environment, University College London, London WC1E 6BT, UK. [7]Institute of Zoology, Zoological Society of London, Regent's Park, London NW1 4RY, UK. [8]Bayreuth Center of Ecology and Environmental Research (BayCEER) & Bayreuth Center of Sport Science (BaySpo), University of Bayreuth, 95447 Bayreuth, Germany. [9]Royal Botanic Gardens Kew, Richmond, Surrey TW9 3AE, UK. [10]Department of Biology, University of Oxford, Oxford OX1 3RB, UK. ✉e-mail: roboke@ceh.ac.uk

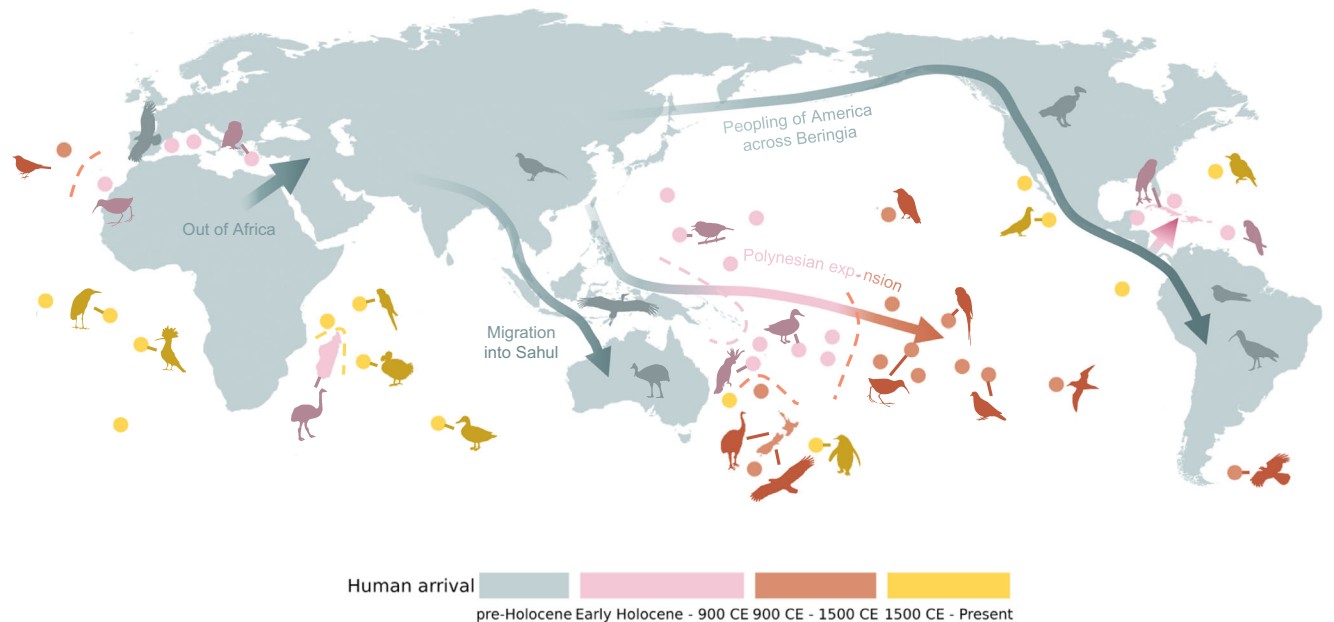

**Fig. 1 | Human colonization and associated bird extinctions.** Human expansion across the planet is classified into four major waves (see legend; Supplementary Fig. 1). Major human dispersal routes are indicated with arrows, and silhouettes show example fossil (pre-Holocene–1500 CE) and observed (1500 CE–Present) bird extinctions. See Supplementary Fig. 1 for additional information and species names. As well as Supplementary Fig. 2 and Supplementary Data 1 for region locations and names. The map is centred on 145°E longitude. The icons are all from PhyloPic.org under Public Domain Dedication 1.0 licenses (see collection https://www.phylopic.org/collections/b2c5ed62-52af-0219-22e1-76a6538ce493). Creator credits: Birgit Lang, FJDegrange, Ferran Sayol, Francesco "Architetto" Rollandin, Juan Carlos Jerí, Mattia Menchetti, Peileppe, Rob Cooke, Sean McCann, Sharon Wegner-Larsen, and Steven Traver.

limits our understanding of avian extinction rates[15] and can substantially underestimate the magnitude of human-driven biodiversity loss[16], with implications for global biodiversity, evolutionary history, and conservation[17–21].

Here, we go beyond previous studies of the Pacific[4,7,10,12] to cover the globe, evaluating the spatial distribution of extinctions, and incorporating undiscovered extinctions, and their uncertainty, into temporal analyses of extinction rate. Hence, we quantify the total magnitude, distribution and rate of bird extinctions worldwide since the Late Pleistocene, including undiscovered extinctions. To estimate undiscovered extinctions, we first modelled fossil bird extinctions across 69 archipelagos (1488 islands; Supplementary Fig. 3) as explained by multiple environmental predictors and the completeness of the fossil record, as indicated by research effort (see Methods). We focus on archipelagos for the model of undiscovered extinctions because ~90% of recorded bird extinctions have occurred on islands[2,22]. We combined these estimates of undiscovered extinctions (archipelagos only) with estimates of fossil and observed extinctions across the globe. Subsequently, we estimated the extinction date for all bird species lost since the Late Pleistocene and inferred extinction rates of birds through time (see Methods). Our findings suggest that ~1300–1500 bird species (~12% of the total) have gone extinct since the Late Pleistocene, more than double the current estimate from the observed and fossil records alone. Moreover, we identify an intense human-driven extinction wave for birds (i.e., caused directly by human activities such as hunting, as well as indirectly through human-associated impacts such as deforestation, fire, and the introduction of invasive species), peaking ~1300 CE with a rate 80 (60–95) times the background extinction rate.

## Results
### Bird extinctions since the Late Pleistocene
Globally, we estimate 1430 (95% credible interval: 1327–1544) bird extinctions since the Late Pleistocene (Fig. 2a). Given current estimates of 10,865 extant bird species, this suggests that 11.6% (10.9–12.4) of all bird species (approximately one in nine) have gone extinct over the last ~126,000 years, with human activities likely contributing to almost all these extinctions. We estimate that 55% of these extinctions are currently undiscovered (788 species; 685–900). Thus, the total number of human-driven bird species extinctions is more than double the current estimate from the observed and fossil records alone (Fig. 2a). For the Pacific we estimate 875 (773–973) total bird extinctions with 554 (450–652) of these undiscovered, compared to 557 (508–605) total extinctions outside the Pacific with 235 (185–281) undiscovered. Hence, we estimate that the Pacific accounts for 61% of total bird extinctions and 70% of undiscovered bird extinctions.

### Bird extinction rate since the Late Pleistocene
Bird extinction rate varies through time, with three major extinction waves since the Late Pleistocene (Fig. 3a and Supplementary Fig. 4). The first, peaking ~840 BCE (Before Common Era; Fig. 3a), was primarily driven by the arrival of people to islands across the Western Pacific (e.g., the Mariana, Tongan and Fijian Islands), as well as the Canary Islands (Fig. 3b). Known extinctions potentially associated with this wave include the Fiji Teal (*Anas* sp.) from Fiji, the New Caledonian Cockatoo (*Cacatua* sp.) from New Caledonia, and the Long-legged Bunting (*Emberiza alcoveri*) from the Canary Islands (Fig. 3b). The second, and most intense extinction wave, peaked ~1300 CE (Fig. 3a) with a maximum extinction rate of 1.6 (1.2–1.9) × $10^{-4}$/species/year, roughly equivalent to ~160 (120–190) extinctions per million species years (E/MSY). This is approximately 80 (60–95) times the background extinction rate, here set to 2 E/MSY following previous assessments[18] (a relatively high estimate of background extinction rate). This wave was driven by the first arrival of people to islands across the Eastern Pacific, especially to Hawaii and the Marquesas Islands, as well as Aotearoa New Zealand (Fig. 3c). The arrival of humans, along with the domestic animals (pigs, dogs, chickens)

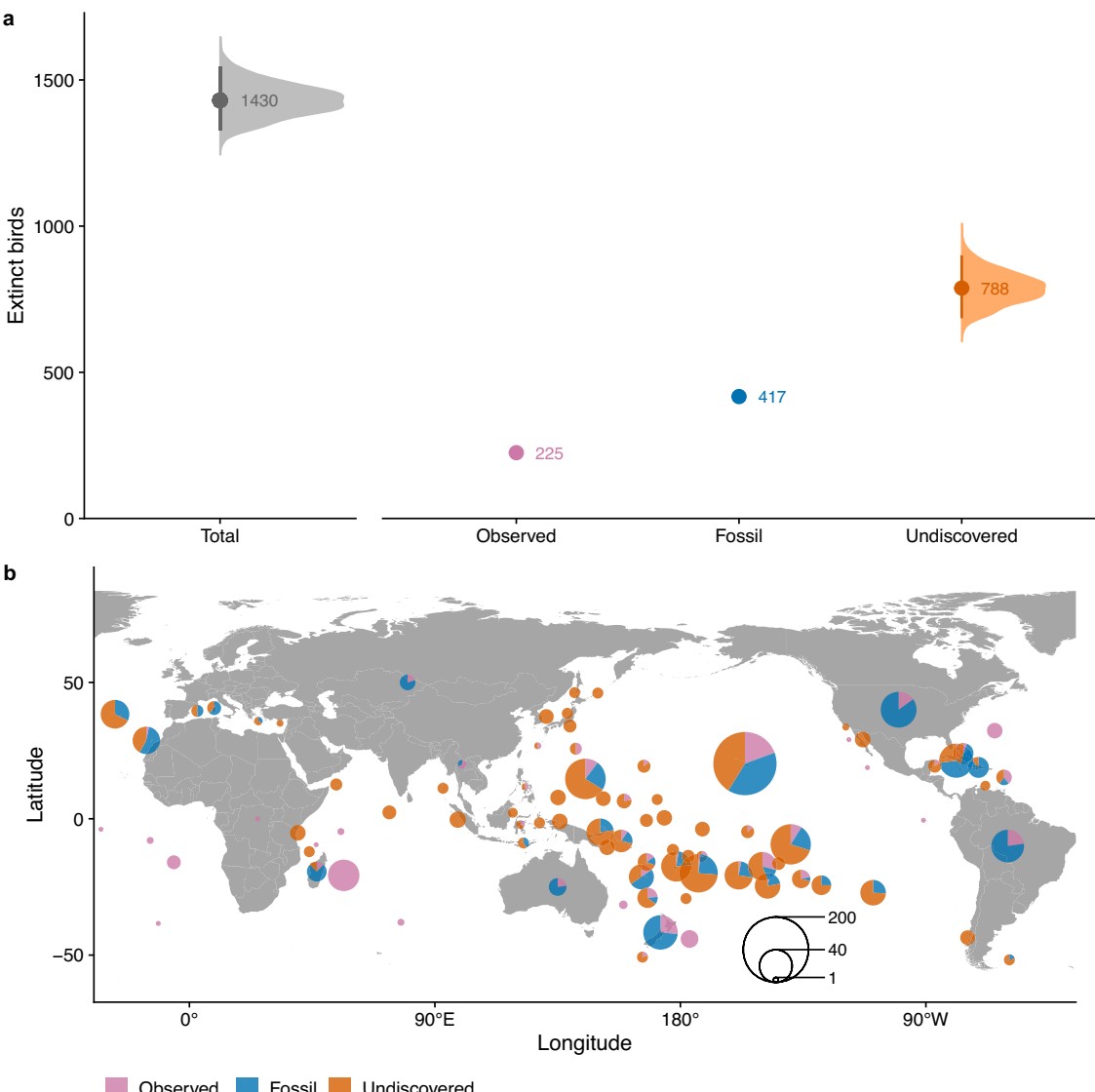

**Fig. 2 | The number and distribution of extinct bird species since the Late Pleistocene. a** The median estimate of total bird extinctions, partitioned into observed (post-1500 CE bird extinctions, plus up to 46 possibly extinct species[22]), fossil, and undiscovered extinctions (those that have not been recorded, and where the species potentially leaves no physical trace, estimated in this study; see Methods). Labelled numbers represent medians. Error bars represent 95% credible intervals (n observed = 1000; n fossil = 1; n undiscovered = 1000). The uncertainty for the observed extinctions is not visible on this scale (lower 95% credible interval = 218, upper 95% credible interval = 230 extinct birds). There is no uncertainty associated with the number of recorded fossils. **b** The distribution of bird extinctions across the globe. The size of the circles represents the total number of extinctions per region; see Supplementary Fig. 2 for region locations. The map is centred on 145°E longitude. Source Data are provided.

and Polynesian Rats (*Rattus exulans*) that were translocated with them, brought habitat transformation and novel threats to the native avifauna, leading to rapid declines of naïve bird species[2,5]. Known extinctions potentially associated with this wave include the High-billed Crow (*Corvus impluviatus*) from Hawaii, Sinoto's Lorikeet (*Vini sinotoi*) from the Marquesas Islands, and nine Moa (Dinornithiformes) species from Aotearoa New Zealand (Fig. 3c). By contrast, the ongoing extinction wave (Fig. 3a), includes extinctions across multiple disparate regions (Fig. 3d), and is driven by the global intensification of human threats, including habitat destruction, direct exploitation, pollution, and invasive non-native species[2,5]. Known extinctions associated with this ongoing wave include 'Āmaui (*Myadestes woahensis*) from Hawaii, Lyall's Wren (*Traversia lyalli*) from Aotearoa New Zealand, and the Colombian Grebe (*Podiceps andinus*) from continental South America (Fig. 3d).

## Discussion

Previous assessments of bird extinction rates that only include observed extinctions (e.g., post-1500 CE[2,13,14,18]) clearly miss major extinction waves (Fig. 3a). Observed high extinction rates have therefore been considered unprecedented[18]. However, at least for birds, earlier extinction waves were greater in magnitude (Fig. 3), as has been hypothesized[10] but not previously demonstrated. In addition, focusing on observed extinctions disregards the fluctuations in extinction rate between waves. Extinctions have typically occurred rapidly after human arrival[2,5,23], but extinction rates have then slowed as vulnerable species were quickly lost and species more resilient to human impacts remained[2] (reflected in Fig. 3a). Thus, the apparent steady increase in extinction rate observed since 1500 CE[13,14,18] (Fig. 3a) may not be appropriate for estimating the true dynamics of the current biodiversity crisis. Instead, here we demonstrate the

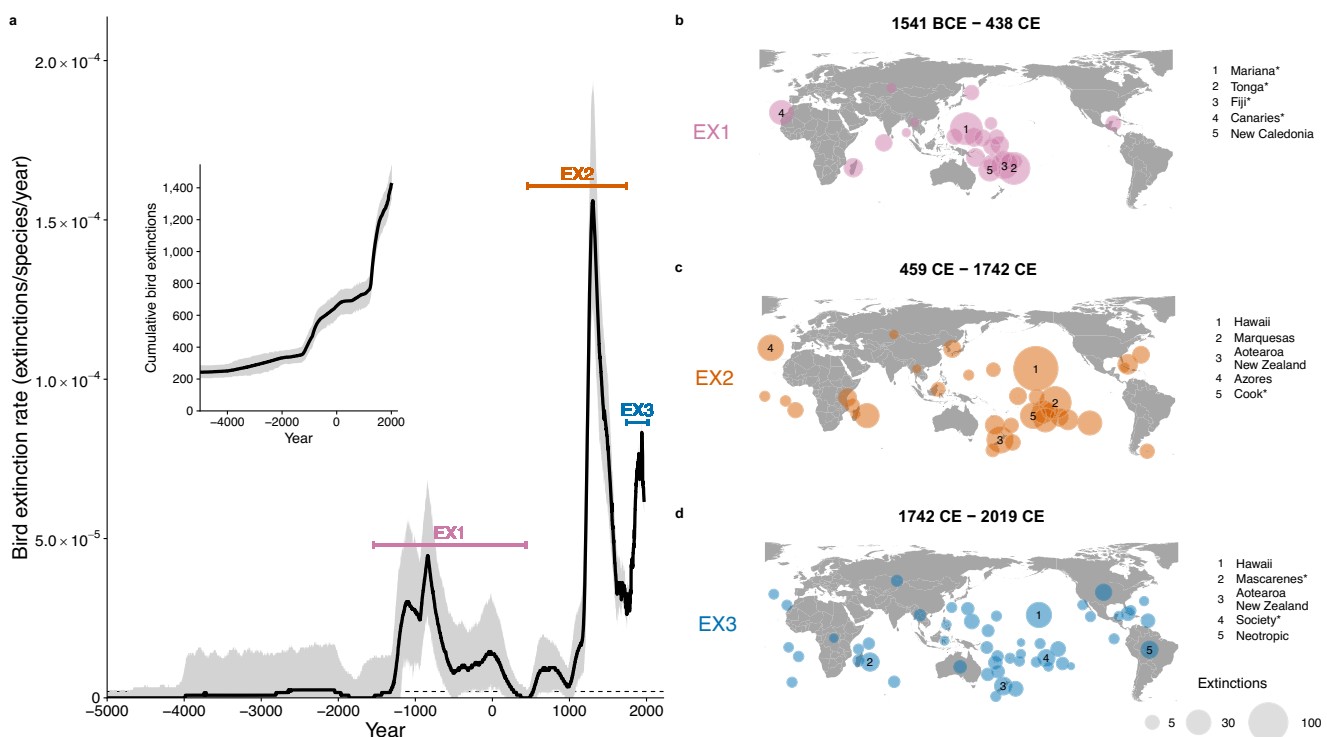

**Fig. 3 | Bird extinction rate through time. a** Median bird extinction rate over the last ~7000 years. **b**–**d** The spatial distribution of three major extinction waves. Most extinctions have occurred over the last ~7000 years (5000 BCE to 2019 CE; see inset and Supplementary Fig. 4 for extinction rate since the Late Pleistocene). Bird extinction rate is the median across 1,000 estimates of rolling means (100-year moving window; Supplementary Fig. 5); envelopes represent 95% credible intervals.

For context, the horizontal dashed line represents a background extinction rate of 2 E/MSY (ref. 18); approximately 0.2 × 10⁻⁵/species/year. Three major extinction waves are labelled, and these are shown spatially in (**b**–**d**). Point size represents bird extinctions and is scaled across the extinction waves. The top five regions are labelled for each wave. The maps are centred on 145°E longitude. Source data are provided.

high spatial and temporal heterogeneity of human-driven extinctions (Fig. 3).

Nevertheless, extinction debts—the projected extinctions of numerous extant species that are yet to occur[24,25]—suggest that we may be moving towards a present-day bird extinction rate even greater than any since the Late Pleistocene[13,26]. Extinction debts can take centuries to be realized[27] and thus today's accumulating extinction debt is still being paid off[19,25]. A recent study estimated an effective (i.e., accounting for the accumulating extinction debt) extinction rate of 2.17 × 10⁻⁴/species/year for the present day[13]; compared to our maximum rate of 1.6 × 10⁻⁴/species/year for ~1300 CE. Moreover, bird extinction rate over the next 100 years has been predicted[26] to reach 6.98 × 10⁻⁴/species/year, a rate that is unprecedented for birds (Fig. 3a). Together these studies predict an additional 226−738 bird extinctions over the next few hundred years[13,26]. If/when these extinctions are realized it would mean that, combined with our results, up to one in six bird species had been driven extinct. The extinction debt, as well as conservation successes[2,13] and the delay in officially listing species as extinct[28], could therefore explain the minor decline in extinction rate we detect since 1939 (Fig. 3a). Hence, in the future, extinctions may be identified and backdated, revising this small decline upwards to a likely increase.

Incomplete knowledge of bird extinctions necessarily underestimates the magnitude of species loss[15,18]. Still, any quantification of undiscovered extinct birds is unavoidably an estimate. Here, we used statistical models and a set of simple, conservative assumptions (in this case, assumptions which should lead to underestimates of the true number). Specifically, we assumed zero undiscovered extinctions since written observational records began and zero undiscovered continental extinctions. We also assumed complete knowledge of the well-studied Aotearoa New Zealand avifauna[7,29], as a way of correcting

for research effort (see Methods). Violations of these assumptions would result in higher overall extinction estimates. Overall, we provide a broad yet precise estimate of undiscovered bird extinctions.

Previous research has estimated the number of bird extinctions across the Pacific[4,7,10,12], with the expectation that most extinctions would be located there (confirmed here as 61% of total bird extinctions). Specifically, these studies use faunal reconstruction[4,10] and variants of mark-recapture[7,12] to estimate bird extinctions. These estimates range from ~800[12], to ~1300[7], to over 2000[4] total Pacific bird extinctions, although definitions of the Pacific vary (i.e., which islands are included/excluded), and which bird groups were studied also differ, making direct comparisons difficult. Still, for the Pacific we estimate 875 (95% credible interval: 773−973) total bird extinctions, which falls within the lower to mid-range of previous estimates[4,7,10,12]. Our Pacific estimate is similar to a previous study that suggested there were more than ~600 but fewer than ~1300 undiscovered extinctions[10]. By contrast, our estimate is lower than a recent estimate of ~1300 for all birds[7], although this study also presents a median estimate for only non-passerine landbirds of 983 (95% credible interval: 731−1332). When considering the uncertainty of both estimates there is strong overlap in the credible intervals, highlighting their general agreement. Our estimate is similar to all previous estimates except for the estimate of over 2000 undiscovered extinctions[4]. This number has been previously described as a likely overestimate[10,30], due to the occurrence of natural events (e.g., volcanism, tsunamis) across some islands assumed to host undiscovered endemic birds that might have prevented colonization or extirpated populations before speciation occurred[10,30,31]. In addition, our estimate of the ratio of undiscovered to discovered extinctions for the Pacific of 0.63 (95% credible interval: 0.59−0.67) is similar to previous estimates of 0.5[12] and 0.67 (95% credible interval: 0.46−0.84)[7]. Thus, our estimate is supported by a range of datasets

and analytical techniques, both within our study (see Methods) and across previous studies[4,7,10,12].

Overall, we estimate that at least -1300–1500 bird species have gone extinct since the Late Pleistocene (-12% of the total). Moreover, we identify an intense extinction wave across the Eastern Pacific around the 14th century, which represents the largest human-driven vertebrate extinction wave reported to date. These extinctions have strong implications for our understanding of avian species richness[2], ecological diversity[19,21] and evolutionary history[20]. The loss of such a large proportion of bird species, with the identities (and therefore ecological roles) of 55% unknown, suggests underappreciated and potentially irreversible ecological impacts, and thus unknown debts in ecosystem functioning[6,21,32,33], particularly across islands, where large native mammals are often absent, and instead birds have adapted to provide important ecosystem functions. Known examples include extinct megaherbviores such as the Elephant Birds (Aepyornithidae) of Madagascar, which influenced plant structure and diversity and ecosystem dynamics[34], extinct aerial predators such as Haast's Eagle (*Hieraaetus moorei*)[35,36], and extinct seed-dispersers such as Seychelles Parakeet (*Psittacula wardi*)[37]. We estimate -800 undiscovered bird extinctions, many of which could also have had key roles in their ecosystems. In addition, bird extinctions can have long evolutionary legacies[20]. These evolutionary impacts may be even greater for archipelagos with large proportions of undiscovered extinctions (e.g., Hawaii), and would amplify the already disproportionate contribution of islands to bird evolutionary diversity[38].

Our study demonstrates the severe and long-term alteration of avifaunas globally[1,3,7], representing a far higher human impact on avian diversity than previously recognized. Urgent protection of the remaining native biotas should constitute a high priority for conservation, to avoid a contemporary extinction wave of even greater magnitude than the prehistoric episodes revealed here.

## Methods
In brief, we estimated the number and timing of bird extinctions since the Late Pleistocene across the globe. Although our outlook was global, we first modelled fossil bird extinctions across archipelagos, as the majority of known bird extinctions have occurred across islands[2,22]. We extrapolated our archipelago model to the research effort of Aotearoa New Zealand, constraining the extrapolations based on upper bounds to obtain estimates of undiscovered bird extinctions across archipelagos. Undiscovered extinctions are those that have not been scientifically recorded: some of these may not have left any physical trace, some may be described in the future[9]. We combined these estimated undiscovered bird extinctions across archipelagos with global fossil (extinct species only known from fossil evidence) and observed (extinct species with written observation records) extinction records to produce a total extinction estimate. We then estimated the extinction rate through time since the Late Pleistocene.

We used R version 4.0.4 (ref. [39]), for all our analyses, except for pre-processing of shapefiles of the archipelagos, which we prepared in ArcGIS 10.4 (ref. [40]) and rasterization of landmasses, which we ran in Python version 2.7.10 (ref. [41]). See https://zenodo.org/records/10014585 for R code summarizing the major analytical steps. We used multiple R packages for data preparation, analysis, and visualization, including arm[42] 1.13-1, broom[43] 1.0.4, cowplot[44] 1.1.1, DHARMa[45] 0.4.6, dplyr[46] 1.1.2, gghalves[47] 0.1.4, ggplot2[48] 3.4.2, HDInterval[49] 0.2.4, hydroGOF[50] 0.4-0, jtools[51] 2.2.1, letsR[52] 4.0, MASS[53] 7.3-53, purrr[54] 1.0.1, raster[55] 3.6-20, readr[56] 2.1.4, rsq[57] 2.5, scales[58] 1.2.1, scatterpie[59] 0.1.8, sf[60] 1.0-13, sp[61] 1.6-0, tibble[62] 3.2.1, tidyr[63] 1.3.0, and zoo[64] 1.8-12.

### Archipelagos and islands
Although most bird species (>80%) live on continents[65], the majority of recorded extinctions (-90%) have been on islands[22]. We therefore acknowledge that the number of continental undiscovered bird extinctions is minor relative to those on islands (i.e., for our analysis we assume zero continental undiscovered extinctions). In addition, post-1500 CE bird extinctions are generally well-documented[2,22], and we therefore assume that the majority of post-1500 CE extinctions are recorded (but see ref. [9] for potential undiscovered extinctions post-1500 CE).

To estimate undiscovered bird extinctions across archipelagos we first extracted geographic and environmental data for 17,883 islands (ref. [66]), we manually added Norfolk Island as it was missing from the dataset. We then identified 1488 islands from this dataset that were settled by humans before written observational records began (i.e., pre-1500 CE) and had the potential to support endemic bird species—islands with area >5 km² (refs. [7, 10]), isolated from the mainland[66–68], and not glaciated[69] during the Last Glacial Maximum (Supplementary Fig. 3). We compiled the date of first human arrival from the literature (Supplementary Data 2). We removed the islands of Cape Verde (humans first arrived 1462 CE[70]) and São Tomé and Príncipe (humans first arrived ca. 1470 CE[71]), as these were first settled close to 1500 CE and have zero fossil extinctions. We therefore conservatively assume that these islands have zero undiscovered extinctions. The area threshold was based on an earlier paper highlighting that only islands >5 km² are likely to support endemic birds[7,10]. We only included islands isolated from the mainland, as islands connected to the mainland are likely to be dominated by continental processes and continental fauna. We compiled previous estimates of island area and connection to the mainland[66], where connection to the mainland was based on global bathymetry data[68] with a sea level of −122 m during the Last Glacial Maximum[66,67]. To identify islands not covered by glaciers during the Last Glacial Maximum we used the prehistoric distribution of ice sheets[69].

We grouped these 1488 islands into 69 archipelagos (Supplementary Fig. 3) according to an existing classification[66], with minor modifications (see isl_arch.csv at https://zenodo.org/records/10014585 for details on islands within archipelagos). Specifically, we aggregated archipelagos that were closely related (e.g., Canary Islands with Madeira), and we disaggregated large archipelagos, as informed by biogeographical (e.g., shared mainland) and ecological (e.g., shared species) similarities between islands, see also ref. [72]. Archipelagos are considered the most-appropriate spatiotemporal insular unit for large-scale biodiversity analyses[73–76]. Here we refer to both true archipelagos (composed of multiple islands) and isolated insular units (single islands; e.g., Cyprus) as archipelagos. All fossil extinct birds were single-archipelago endemics except for nine species (2%): *Alopecoenas nui, Caracara creightoni, Gallinago kakuki, Megapodius alimentum, Megapodius molistructor, Tyto noeli, Tyto pollens, Vini sinotoi,* and *Vini vidivici,* which we assigned to their primary archipelago, based on island area. We therefore assume that all undiscovered extinctions were single-archipelago endemic species.

We constructed shapefiles in ArcGIS 10.4 (ref. [40]) for each of the archipelagos from the database of global administrative areas (GADM), version 3.6 (gadm.org/data.html).

### Data preparation
To estimate the number of undiscovered extinct species (not yet discovered or left no trace) we first modelled the number of fossil extinct bird species across the focal archipelagos. We calculated the number of fossil extinct birds for each of the archipelagos from a recently compiled database[72].

We also compiled data for 14 predictors (aggregating island-scale measurements to the archipelago-scale where applicable; Supplementary Data 3), underpinned by biological a priori expectations. Although soil chemistry is known to affect fossil preservation[77,78], suitable data at the global scale are too imprecise and the influence of soil properties on preservation is often local and

complex[78]. We manually added data for Norfolk Island, which was missing from[66], collating data from WorldClim[79], SRTM30 (ref. 80), P. Weigelt (pers. comm., Feb 2020), and adjusting the Lord Howe Island estimate of archipelago plant richness based on the species-area relationship[81,82]. Here we outline the predictors and our respective a priori expectations.

1. Research effort. We expect greater research effort, as approximated by the number of publications referring to biological fossils, to relate to greater probability of discovering bird fossils.

   We estimated research effort as the number of eligible publications referring to biological fossils per archipelago. We followed the Preferred Reporting Items for Systematic Reviews and Meta-Analysis (PRISMA) guidelines[83]. To identify publications, we used Scopus (www.scopus.com; accessed 2 March 2020) with the search query string:

   (TITLE-ABS-KEY ("Archipelago_names" OR "Island_names")
   AND TITLE-ABS-KEY (fossil)
   AND NOT TITLE-ABS-KEY ("fossil fuel" OR "fossil reef" OR "fossil coral" OR "fossil beach*"))

   The phrases "fossil fuel", "fossil reef", "fossil coral" or "fossil beach" were excluded to ensure the search was focused on terrestrial, biological references. We extracted archipelago names primarily from[66], with manual additions based on the literature. We also included up to six alternative (sub-)archipelago names that might be referred to in the literature. We extracted island names from[66] and the literature, and included alternative island names where applicable. We included both English and regional archipelago/island names where possible, e.g., Flores Island and Pulau Flores, and Scopus accounts for accented characters, e.g., Yucatan and Yucatán. We ran the search for each of the archipelagos. However, for Aotearoa New Zealand we removed all publications after 2009 (the date of discovery of the last fossil bird species). Thus, our measure of research effort reflects the research effort to date for all archipelagos (potentially incomplete avifaunas), except for Aotearoa New Zealand, where it reflects the research effort required to discover the complete avifauna[29]. Our search resulted in 4938 total publications (Supplementary Fig. 6). We removed 25 within-archipelago duplicates and screened the remaining publications for eligibility. We screened all publication titles, and when eligibility was uncertain, we read the abstracts and keywords. We excluded publications if the main study location was not the focal archipelago (this occurred where multiple archipelagos had the same island names, e.g. South Island), if the publication focused on marine fossil samples (e.g., benthic samples, fossil coral), if they did not focus on biological fossils (e.g., fossil fuels, volcanic fossils) and/ or if they referred to living fossils—resulting in 2139 eligible publications (Supplementary Fig. 6).

2. Isolation distance. Isolation is a major driver of island biodiversity, as isolation facilitates in-situ speciation, which leads to high levels of endemicity in islands (despite islands tending to have lower species richness overall)[31,84]. Isolation can increase extinction susceptibility, because the avifaunas of isolated archipelagos are likely to have evolved for longer in the absence of mainland predators, and so their species are more likely to react naïvely to both human hunters and exotic mammalian predators when they arrive[3,85,86]. Archipelagos that are more isolated are likely to have lower rates of colonization, leading to fewer species overall, but higher rates of endemism[87], and thus greater probability of global, rather than regional, extinction. A classic example of this phenomenon is Hawaii[88]. Isolation distance also relates to the overwater distance from the mainland required to reach the archipelago, which acts as a dispersal filter, modifying community composition[7].

3. Surrounding landmass. As for isolation distance, surrounding landmass relates to species naïvety to predators. Unlike isolation distance, surrounding landmass accounts for coastline shape of large landmasses and therefore relates more closely to colonization pressure[87], although surrounding landmass does not as closely represent the overwater distance required to reach the archipelago (i.e., the dispersal filter).

4. Elevation. Mountainous regions generally have limited human accessibility, providing refuge from human hunters, but also impeding attempts to discover fossils[7,89–91]. Elevation also relates to topographic complexity and environmental heterogeneity, both of which can affect speciation and extinction processes.

5. Temperature. Climatic factors, such as temperature, are key drivers of ecosystem processes and vegetation structure[66], therefore influencing speciation, colonization, and extinction. Moreover, climatic factors can mediate human impacts on archipelagos; for example, dry islands are more likely than wet islands to end up deforested[92].

6. Precipitation. As for temperature.

7. Temperature variability. As for temperature.

8. Precipitation variability. As for temperature.

9. Archipelago plant richness. Plant richness relates to the productivity and niche diversity within an archipelago and therefore the support for higher trophic levels such as birds[93]. Hence, archipelago plant richness relates to the number of bird, and therefore extinct bird, species supported within an archipelago.

10. Total area. Area is a major driver of island biodiversity[31,84]. Archipelagos with greater total area support larger (meta-)populations of bird species, which are thus less prone to extinction[3,31]. Archipelagos with greater area are also more likely to provide refuge from extinction pressures[3,7]. Archipelagos with greater area likely require more sampling effort to discover fossil specimens. Thus, archipelago area can affect both the number, and detection, of extinct birds.

11. Standard deviation (SD) in area. SD area relates to the disparity in island size within an archipelago and is therefore associated with meta-population processes, such as source-sink dynamics[94]. In addition, SD area captures differences in structure between archipelagos, e.g., archipelagos with a few large islands vs many small islands.

12. Native rodents. The impact of (human-introduced) exotic predators, particularly rats and mice, is reduced on islands that possess native rodents, likely because these avifaunas are less naïve to the effects of nest predation[95,96]. Thus, insufficient eco-evolutionary exposure (i.e., absence of native rodents) likely drives avian naïvety towards exotic predators[95], and therefore relates to extinction risk. Approximately a third of the focal archipelagos have native rodents[97] and hence likely reduced extinction risk from exotic predators.

13. Human arrival. Archipelagos colonized long ago have had more time for fossil specimens to degrade, deteriorate and disappear[98]. In addition, the timing of peopling of an archipelago could affect the magnitude and structure of human pressures[6,99,100]. Indeed, human arrival could reflect changes in agricultural practice and land modification (widespread land clearance and domesticated plants are more often associated with later human settlements)[99], changes in hunting preferences, technologies and efficiency (e.g., use of bows, arrows, and spears; fishing capabilities)[100,101], changes in the number of human introductions and commensals (e.g., pigs, rats)[2,5,6], and changes in human behaviour and culture (e.g., communication/language, long-distance trade)[99,100].

14. Endemic birds. The number of (remaining) endemic bird species likely relates to evolutionary processes, such as dispersal and speciation, and therefore to general avian diversity. Hence, the

number of endemic birds likely relates to the number of extinct birds.

We quantified the number of endemic bird species as the number of extant and observed extinct (i.e., those extant at 1500 CE) endemic birds per archipelago. To identify endemic birds we used expert-based species' range maps for 11,165 species[102]. We restricted species ranges to only include areas where they were listed as native or reintroduced (i.e., we excluded introduced or vagrant areas, etc.) and where their seasonality was listed as resident or breeding season (i.e., we excluded passage areas, etc.). For 344 seabird species, we restricted their ranges further to only breeding season areas (i.e., removing areas where seabirds are resident at sea). We defined seabirds taxonomically as groups where all species feed at sea, either nearshore or offshore (Spheniscifromes, Procellariiformes, Phaethontiformes, Pelecaniformes [family Pelecanidae], Suliformes [families Fregatidae, Phalacrocoracidae and Sulidae] and Charadriiformes [families Stercorariidae, Laridae and Alcidae][103]). We then constructed convex hulls for the archipelagos and buffered them by 4000 m (the shortest distance between any archipelago and the mainland = 4200 m) to account for any spatial mismatches between the species' ranges and the archipelagos, and to ensure that small islands and islets (<5 km²; small islands and islets might be occupied by satellite bird populations but are unlikely to be large enough to support endemic species[10]) were included. We then calculated the number of species endemic to the archipelago by summing the number of species that were found within each buffered archipelago convex hull (st_covered_by function from the sf package[60]).

Total area was strongly correlated with SD area (Pearson $r = 0.91$, df = 67, $P < 0.001$), so we included only total area in the regression models, but total area can be interpreted in light of their covariation. Isolation distance and surrounding landmass were also strongly correlated (Pearson $r = -0.84$, df = 67, $P < 0.001$), so we included only isolation distance in the models, but again this means that the estimated effect can be interpreted as the effect of isolation distance or surrounding landmass. This resulted in 12 suitable predictors (Supplementary Data 3).

## Model of fossil extinct birds

First, we fit a linear model (LM) of log(fossil extinct birds + 1) as explained by the main effects of the 12 suitable predictors (Supplementary Data 3) and the interaction between total area and research effort. We transformed the predictors where it improved the distribution of the residuals and/or the stability of the variance (Supplementary Data 3). All predictors were centred and scaled to zero mean and unit variance, to improve the interpretability of regression coefficients[104]. All variance inflation factors were <7 indicating acceptable levels of multicollinearity[105]. We found no evidence of spatial autocorrelation (Moran's tests for one to 20 neighbours). To quantify model uncertainty we simulated >1000 posterior estimates[106], via the sim function in the arm package[42]. For similar applications see refs. [107],[108].

Transformation of count data for modelling purposes is often not recommended, due to the expected poorer performance of LMs compared to generalized linear models (GLMs)[109]. Hence, we also fit a quasipoisson (the data were overdispersed) GLM. Although a GLM might seem preferable compared to LM[109], we were concerned that the strongly asymmetrical errors of the GLM had the potential to generate upwards bias; hence the creation of the LM as well.

To test the performance of the LM vs GLM we used leave-one-out cross-validation. We evaluated the predictive performance of the models (comparing the observed to the predicted numbers of fossil bird extinctions) with three goodness of fit measures using the gof function (hydroGOF package[50]): Spearman's rank correlation, cross-validated R², and percent bias. Spearman's rank correlation measured the model's ability to correctly rank archipelagos according to their number of extinctions. Cross-validated R² gave the proportion of the variance of the observed extinctions that is predictable from the estimated extinctions, highlighting the model's efficiency. Percent bias measured the average tendency of the estimated extinctions to be larger or smaller than the observed extinctions. Overall, the LM had better predictive performance than the GLM, with similar Spearman's rank correlation (LM = 0.73; GLM = 0.74), greater R² (LM = 0.42; GLM = 0.14), and similar absolute percent bias (LM = -35; GLM = +29). Crucially, the GLM showed positive percent bias—the GLM overestimates bird extinctions—likely driven by the strongly asymmetrical errors of the GLM. Overestimation is particularly concerning for extrapolative approaches (see below), as extrapolation can amplify these overestimations. For instance, after extrapolation, 23% of the GLM estimates were greater than 1000 undiscovered extinctions for a single archipelago, a biologically implausible estimate. Hence, the LM shows both better performance and more conservative extinction estimates, so we chose to take the LM forward for further analysis.

The LM explained a substantial proportion of the variance in fossil bird extinctions (multiple $R^2 = 0.68$; adjusted $R^2 = 0.60$). The strongest predictor was research effort (partial $R^2 = 0.26$), followed by human arrival (partial $R^2 = 0.13$), and then isolation distance, elevation, temperature and native rodents (all partial $R^2 = 0.03$) (Supplementary Data 4). See below for further validation, performance testing, and constraints applied to the LM.

## Extrapolation

Next, we used the LM to extrapolate the number of extinct birds for each archipelago (Fig. 4). We extrapolated based on the research effort required to produce a complete inventory of extinct bird species for Aotearoa New Zealand. In other words, how many extinct birds would we expect if all archipelagos had the same research effort as Aotearoa New Zealand needed to discover their complete avifauna? An advantage of this extrapolative approach is that the total estimated number of extinct birds per archipelago must necessarily be greater than or equal to the observed number (biologically realistic), whereas other approaches (e.g., prediction-based approaches) can estimate fewer extinctions than observed (i.e., negative extinctions; biologically impossible). In addition, for Aotearoa New Zealand the estimated value is necessarily equal to the observed value. Thus, our estimates are benchmarked on Aotearoa New Zealand. We used Aotearoa New Zealand as our reference archipelago as extinction rates here are not subject to the same uncertainties associated with incomplete detection elsewhere[7,29]. Specifically, Aotearoa New Zealand was only recently colonized by humans (ca. 1250 CE[110]), contains abundant, well-preserved remains of the Holocene bird assemblage (all modern native species have been found as remains[111]) and is the only place in the world where the pre-human avifauna is believed to have been comprehensively reconstructed[29,85,112–115]. Thus, Aotearoa New Zealand is the only reasonable point of reference for these analyses. In addition, Aotearoa New Zealand had a research effort of 343 eligible publications by 2009 when all extinct species were described, while the next most researched archipelago, Hispaniola*, had a research effort of 270. Thus, the analyses could also be interpreted as extrapolating to the maximum observed research effort (Fig. 4). For each archipelago we extrapolated the total number of extinct birds using the LM and the difference in research effort (and the scaling of increasing research effort with increasing total area, fit as an interaction term in the model) as compared to Aotearoa New Zealand, with all else being equal. We calculated total number of extinct birds for each of the >1000 posterior estimates from the LM. We back-transformed the extrapolations to the natural scale and then subtracted the number of known fossil extinct birds from the total to obtain estimates of the

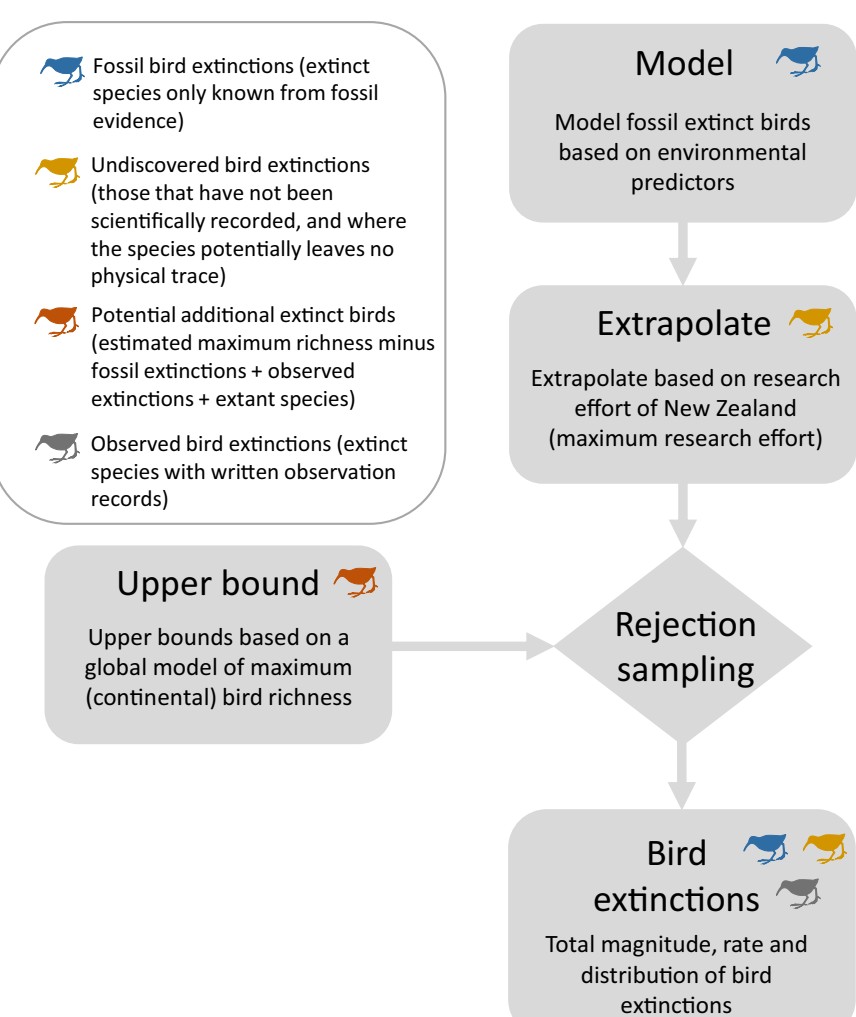

**Fig. 4 | Overview of major methodological steps.** Diagram of the major methodological steps involved in the analysis, highlighting how the different categories of bird extinction were used and estimated.

number of undiscovered extinct birds. We converted our extrapolations to species numbers (integers) using probability-based rounding. For instance, the value of 1.7 had a 0.7 probability of being 2 extinctions and a 0.3 probability of being 1 extinction.

We have described our estimates of undiscovered extinct birds as extrapolations, as we extrapolated outside the observed data for research effort. Yet, for the remaining predictors in the model, the extrapolations generally fell within the range of observed data (i.e., interpolations; Supplementary Fig. 7).

**Archipelago upper bounds**
To test and ensure our estimates of the number of undiscovered bird extinctions were biologically realistic we calculated archipelago-specific upper bounds (Fig. 4).

We calculated upper bounds to constrain extremely high values—maximum estimates, above which predictions would be biologically unrealistic. We based our upper bounds on island biogeography theory, which states that continental bird diversity is greater than island bird diversity per unit area, due to the smaller system size (higher extinction) and greater geographical isolation (reduced immigration) of island systems[31,38,73,116]. We therefore quantified the maximum bird diversity of the archipelagos as if they were continental, producing estimates of maximum potential bird diversity.

To quantify the upper bounds, we first calculated the number of extant and observed extinct (i.e., extant at 1500 CE) bird species at 1°

resolution[117] across continental areas. We used range maps for 11,165 bird species, prepared the same as for the endemic birds (see above; except instead of using the st_covered_by function [a species' range is wholly within an archipelago, i.e., endemic] we used st_intersects [any of the range intersects with the archipelago][60]). We converted the range maps into a global presence-absence raster using a polygon-to-grid procedure (lets.presab function; letsR package[52]) and summed the number of bird species per grid cell to obtain bird species richness.

We then compiled five environmental predictors at 1° resolution equal-area projection across continental areas: elevation, temperature, precipitation, temperature variability, and precipitation variability (transformed where applicable; Supplementary Data 5). We extracted elevation from SRTM30[80], while the bioclimatic predictors were extracted from WorldClim[79]. We excluded grid cells with missing predictor data as well as grid cells within the arctic or Antarctic circles, due to the transient dynamics of birds in these regions[118]— resulting in 10,600 grid cells.

We modelled bird species richness as explained by the five environmental predictors with a quasipoisson GLM. To reduce residual spatial autocorrelation we also included a residuals autocovariate term[119], built from the residuals of an initial non-spatial GLM, and an optimized spatial neighbourhood structure (first-order neighbourhood was optimal based on Moran's *I*). To quantify uncertainty we simulated >1000 posterior estimates[106] from the spatial GLM, using the sim function (arm package[42]).

We then predicted the species richness of the 68 focal archipelagos (excluding Aotearoa New Zealand) as if they were continental (median predicted continental richness across grid cells per archipelago). We scaled the grid-based continental richness estimates to the total area of the archipelagos using the canonical species-area relationship[81,82]. We then subtracted the number of fossil extinctions, observed extinctions, and extant bird species of an archipelago from the continental richness to obtain estimates of the potential additional species that an archipelago could support (Fig. 4). Where continental richness fell below recorded species richness, we set the number of potential additional species to zero (i.e., recorded species richness is at or close to the predicted continental maximum). We used these estimates of potential additional species as upper bounds, above which predictions would be biologically unrealistic.

The upper bound spatial model explained a substantial proportion of the variance in continental bird species richness (McFadden pseudo-$R^2$ = 0.86; Supplementary Data 6). The upper bound model also showed good cross-validated predictive performance (Spearman's rank correlation = 0.99, $R^2$ = 0.93, percent bias = 0.00).

### Global lower bound

Note, we also calculated a global lower bound of the number of describable species by fitting a Bayesian logistic function to a species-description curve[120] using Markov Chain Monte Carlo (MCMC) estimation. Virtually all of our estimates of undiscovered extinctions were greater than the lower bound, so, for simplicity, we do not take the lower bound further, but methodological details are provided here.

We calculated a global lower bound, a minimum estimate, below which predictions would be biologically unrealistic. To characterize the lower bound, we estimated the number of describable (those species that have been described from fossils and those that are likely to be described in the future, based on current efforts) fossil extinct bird species. This is necessarily a lower bound, as it does not account for those species that are unlikely to ever be discovered, e.g., species that have gone extinct without fossilization or species that require intensive research effort to be discovered[3,7,10].

To quantify the number of describable species we fit a logistic function to a species-description curve[120], based on fossil bird extinctions and the date that they were described[72]. We then modelled the number of described species at time $t$ as samples from a normal density:

$$N_t \sim N(\mu, \sigma) \tag{1}$$

with mean $\mu_t$, following a logistic function and variance estimated from the data. We fit the Bayesian model using Markov Chain Monte Carlo (MCMC) estimation, with a sampling frequency of 5000, a burn-in period of 100,000 iterations and flat unbounded priors, except for the midpoint, which we forced to be in the observed time window (1843 CE–2017 CE). We therefore assume that the description of ≥50% of all prehistoric extinct birds occurs sometime after 1843 CE and before 2017 CE, which is a reasonable assumption given that there are fewer species being described per author than previously, suggesting it is already harder to discover new species[121,122]. To get the number of describable species, we obtained posterior estimates of the maximum of the logistic (excluding the burn-in period); we then subtracted the number of described species (283 species) to estimate the number of undescribed species (Fig. 5).

### Rejection sampling

We used rejection sampling to integrate the estimates of undiscovered extinct birds from the LM with the archipelago upper bounds (Fig. 4). We rejected estimates of undiscovered bird extinctions where the archipelago had a value greater than their respective upper bound estimate. We rejected relatively few (median = 0.2% rejected; mean = 12% rejected) estimates per archipelago based on the upper bounds (Supplementary Figs. 8, 9). Archipelagos with high rejection rates generally had similar recorded bird species richness to continental richness, i.e., recorded bird richness is at or close to maximum, e.g., Sado*, Derawan*, Yucatan* (Supplementary Fig. 8).

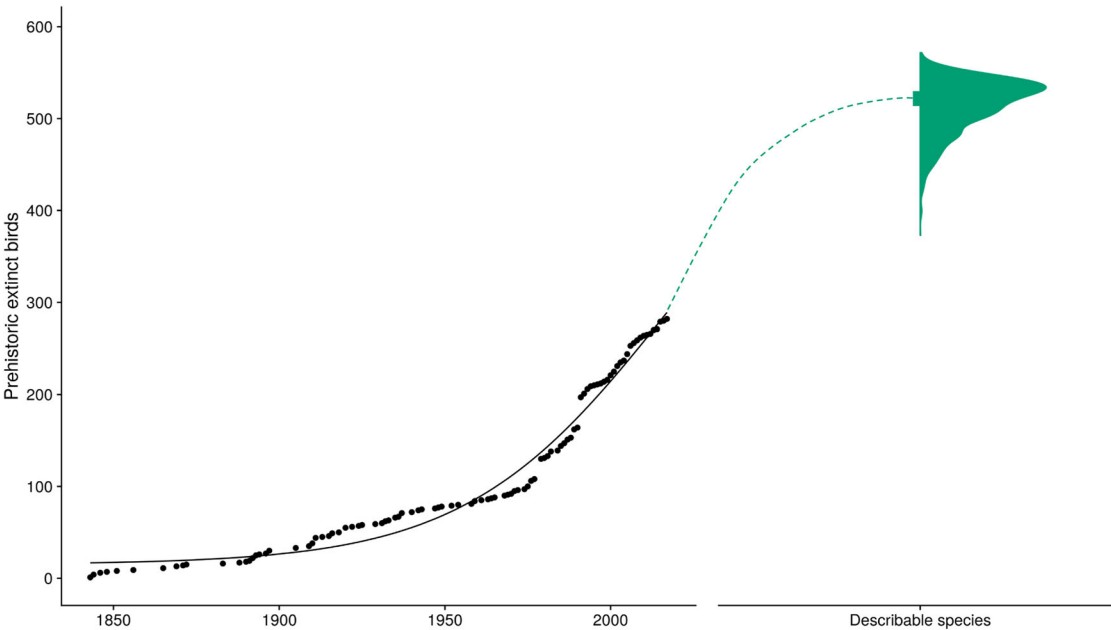

**Fig. 5 | Lower bound estimation.** Observed species-description curve between 1843 CE and 2017 CE, and the estimated describable fossil extinct bird species. The species-description curve shows fossil extinct birds (black points) and the description year, as well as the modelled logistic curve (black line). A green dashed line (not statistically derived or time explicit) is also included, visualizing the link to a sample of 1000 estimates of describable birds (green half violin, with a square representing the median = 522). Describable birds consist of both described (283 species) and undescribed fossil extinct birds (median = 239). Source data are provided.

## Alternative model of fossil extinct birds

As an alternative modelling approach to the LM model described above, we also implemented a white noise model with multiple predictors using a Bayesian framework, where all parameters are sampled from their posterior distribution using a MCMC algorithm. The mean of the white noise process ($\mu_x$) changes as a linear function of multiple predictors ($P$) and their estimated effect sizes ($e$), following the linear formula:

$$\mu_x = \mu_0 + \sum_{i=0}^{n} e_i \times P_i \qquad (2)$$

We used the same predictors ($n = 12$) in the white noise linear model as for the LM above (Supplementary Data 3). We also modelled explicitly the interaction between the two predictors total area and research effort with its own effect size parameter ($e_{(n+1)} \times P_{(total\ area)} \times P_{(research\ effort)}$), which was added to the linear formula above. The variation (noise) around the mean of the white noise process is captured by a separate parameter ($\sigma$), describing the standard deviation of the normal distribution that is centred in $\mu_x$. During the MCMC we sampled the parameter estimates for $\mu_0$, $\sigma$, and the effect sizes ($e_i$) for all predictors, including the interaction term. The parameter $\mu_0$ can be interpreted as describing the expected baseline number of extinctions if all predictor values were 0. As the likelihood formula for the MCMC we applied the probability density function of the normal distribution. As priors we applied for $\mu_0$ a uniform prior with the limits 0–200, for $\sigma$ a gamma prior with a shape of 1 and a rate of 0.01, and a normal prior on all effect size parameters with a mean of 0 and an exponential hyperprior on the standard-deviation with a rate of 1.

The MCMC was run for 300,000,000 iterations, sampling every 500 iterations. Although the MCMC converged already after 1,000,000 iterations, we continued the MCMC for this many iterations to ensure sufficient parameter samples for the following rejection sampling (after excluding 10% as burn-in).

Again, when comparing the predicted to the observed bird extinctions for the archipelagos, the white noise linear model (Spearman's rank correlation = 0.65; $R^2 = 0.59$; percent bias = +11) outperformed an exponential version of the white noise model (Spearman's rank correlation = 0.60; $R^2 = 0.07$; percent bias = +207; some extreme estimates, i.e., >1000 undiscovered extinctions for a single archipelago).

We applied the same extrapolative approach as above to obtain estimates of undiscovered extinctions for the white noise linear model. We also applied the same process of rejection sampling. Again, we rejected relatively few (median = 0% rejected; mean = 11% rejected) estimates per archipelago based on the upper bounds.

The results for the two modelling approaches (LM and white noise linear model) show moderate correlation (Spearman's rank correlation = 0.5), although the total estimates are very similar (LM = 789 total undiscovered extinctions; white noise linear model = 935 total undiscovered extinctions). In the end, we decided to only take the LM forward for further analysis, as it was more conservative in terms of estimated extinctions, easier to interpret, and seemed to capture more of the differences between archipelagos (greater variance). Still, the similarity of the total estimates between the two modelling approaches gave us confidence in the validity of our extinction estimates.

## Comparison to previous data

To further test the validity of our estimations we also compared them to previous estimates across the Pacific[4,7,10,12]. Although direct comparison is problematic, due to differing geographic and/or species inclusion criteria (e.g., different definitions of the Pacific, and/or different bird inclusion–single island endemics, only landbirds, etc.). For our estimate, we included archipelagos to match the coverage of ref. 10, representing Micronesia, Melanesia and Polynesia.

## Extinction rate

After calculating the number of bird extinctions across the globe, we estimated the timing and rate of extinctions. To calculate the extinction rate, we first estimated extinction dates for all species.

**Fossil.** For all fossil bird extinctions we estimated extinction dates based on truncated exponential decays, following first human arrival. Hence we assume that the most extinction-prone species will be lost rapidly after human arrival and the rate of extinction will then slow[8]. For all archipelagos, except Madagascar, we modelled 75% of extinctions to occur within 200 years of first human arrival (half-life of 100 years), based on the well-studied extinction chronology of Aotearoa New Zealand[23,123,124]. We truncated the exponential distributions so that 100% of extinctions occurred at the 90th quantile, i.e., 100% of extinctions occurred within 332 years of first human arrival–to prevent predicting extinction dates far outside the window of initial human impact. We therefore assume that the majority of initial human disturbance and human-driven extinctions are likely to have occurred within a few centuries[23,123,124]. This estimate is likely conservative across the focal archipelagos, as most are smaller in area than Aotearoa New Zealand; therefore, it is likely that all else being equal, humans would have colonized–and impacted–more rapidly across these smaller archipelagos.

Madagascar, on the other hand, is much larger than Aotearoa New Zealand (2.2 times larger) and is often described as an island-continent, as it is influenced by both island-like and continent-like processes[125,126]. We therefore applied a longer extinction window to Madagascar compared to the other archipelagos. We modelled 75% of extinctions to occur within 2000 years of first human arrival (half-life of 1000 years), based on the well-studied extinction chronology of North American Pleistocene mammals[127]. In other words, we assume that the extinction chronology of Madagascar is more similar to the timing of continental extinctions than to the timing of island extinctions (where Aotearoa New Zealand and North American Pleistocene mammals are the only available, suitable extinction chronologies). For Madagascar, we truncated the exponential distribution so that no extinctions occurred post-1950 (i.e., 100% of extinctions occurred at the 75th quantile).

There were also 98 continental fossil bird extinctions, which we assigned to their respective biogeographic realms[128]. For the Australasia, Nearctic and Neotropic realms, which have better established and later human arrival dates[129,130], compared to the Indo-Malay and Palearctic[131], we estimated extinction dates based on truncated exponential decays after first human arrival. We used a Clovis-culture human arrival date for the Nearctic[130], even though there is evidence of earlier human arrival[132,133], as no consensus has been reached among archaeologists about the date of initial human arrival to the Americas (and evidence of human arrival south of the ice sheets is fragmented), but all agree that human populations were distributed across the North American continent by the Clovis period[134], implying greater human influence at this point. We modelled 75% of extinctions to occur within 2000 years of first human arrival (half-life of 1000 years), based on the extinction chronology of North American Pleistocene mammals[127], and truncated the exponential distributions so that 100% of extinctions occurred at the 90th quantile.

When human arrival date was uncertain for a region (archipelago or continent), we randomly sampled (for each of the 1000 extinction estimates) from a uniform distribution based on the arrival date bounds.

For the Indo-Malay and Palearctic realms we randomly estimated extinction dates per species from uniform distributions between the minimum age of the fossil and 1500 CE, i.e., the species went extinct

sometime after the fossil was deposited and before preserved written observational records began. We used a different approach for these realms as they have much earlier and more complex human arrival sequences[131].

**Observed.** For observed extinctions we used the date of the last reliable or confirmed record[22,72,135], where available. For 17 species, date of the last reliable or confirmed record was unavailable. For these species, we estimated extinction dates from truncated exponential distributions (half-life of 100 years), following first human arrival (see above).

**Observed possibly extinct.** We also accounted for possibly extinct species[22]. Species are only classified as Extinct (or Extinct in the Wild) on the IUCN Red List if there is no reasonable doubt that the last individual has died[136]. This strict definition of extinction aims to reduce the Romeo error—when conservationists give up on a species prematurely[137]; but failing to recognize extinctions leads to underestimates of extinction rates[22,28]. Thus, we incorporated 46 possibly extinct species that have been previously identified[22]. To incorporate the possibly extinct species, we used the reported mean extinction probability[22]. Then, for each of the 1000 extinction estimates, we randomly sampled from a binomial distribution based on the species' probability of extinction—resulting in ≤46 additional extinctions per estimate. For these species we used the date of the last reliable or confirmed record as the extinction date[22,135].

**Overall extinction rate.** We generated 1000 potential extinction chronologies for each of the archipelagos, accounting for variation in the number of bird extinctions (variation in the number of undiscovered extinctions and possibly extinct species), timing of human arrival (uncertainty associated with human arrival dates) and exponential decay (randomization of exponential extinction sequences). We calculated extinction rate as the number of extinctions at time $t$ divided by the number extant at time $t$ (extinctions/species/year) for the period 124,050 BCE–2019 CE, with 10,865 extant birds at present. We then calculated a rolling mean of yearly extinction rate with a window of 100 years. We summarized extinction rate as the mean across the 1000 extinction estimates.

We report extinction rate on the natural scale (extinctions/species/year), although others have advocated for the scale extinctions per million species years (E/MSY)[8]. Our extinctions/species/year estimates were therefore very roughly converted to E/MSY by multiplying by one million[13], however see[13] for a discussion of the issues with E/MSY, including converting per-year estimates of extinction to E/MSY and comparing E/MSY across different numbers of species. Thus, following ref.[13] we focus on extinctions/species/year estimates, while E/MSY is only used to put our estimates in the context of background extinction rates[8,18,138].

Although the exact causes of bird extinctions are debated[1,6,23,139], here we assume humans contributed (directly or indirectly) to all bird extinctions, supported by multiple lines of evidence[2,3,5,6,85,124,140,141]. There are few meaningful exceptions to this assumption—the Maltese Giant Swan (*Cygnus falconeri*) is one likely natural extinction[142]. Excluding these species from our analyses would lead to incorrect extinction rate estimates, plus the impact of these natural extinctions on the total extinctions is minimal; thus, we included them in all analyses.

**Reporting summary**
Further information on research design is available in the Nature Portfolio Reporting Summary linked to this article.

## Data availability
All prepared and processed data are available at https://zenodo.org/records/10014585 (doi: 10.5281/zenodo.10014585). Raw data on island characteristics (e.g., island area, island precipitation)[66] are available from https://doi.org/10.5061/dryad.fv94v. Shapefiles of the archipelagos are available from the database of global administrative areas (GADM) https://gadm.org/data.html. Information on fossil extinct birds[72] is available from https://doi.org/10.5061/dryad.s1rn8pk66. Raw information on native rodents[97] is available from https://doi.org/10.5281/zenodo.1250504. Shapefiles of bird distributions are available from http://datazone.birdlife.org/species/requestdis. Global elevation data[80] are available from https://srtm.csi.cgiar.org/srtmdata/, while WorldClim[79] data are available from https://www.worldclim.org/data/worldclim21.html or via the getData function from the raster R package[55]. Bird extinction probabilities and the date of recent bird extinctions[22] are available from https://data.mendeley.com/datasets/vvjhpmyxb4/1. Source data are also provided with this paper. Specifically, the source data underlying Figs. 2, 3 and 5, and Supplementary Figs. 4, 5, 7–9 are provided. Source data are provided with this paper.

## Code availability
R code to reproduce the analyses is available at https://zenodo.org/records/10014585 (https://doi.org/10.5281/zenodo.10014585).

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

## Acknowledgements

The authors thank P. Weigelt for providing environmental data for Norfolk Island, as well as D. Silvestro and R. Duncan for help with the analyses. We also thank R. Smith and T. Matthews for comments on the manuscript. F.S. received support from "la Caixa" Foundation (ID 100010434) through a Junior Leader program (fellowship code LCF/BQ/PI23/11970019). T.A. received financial support from a SciLifeLab & Wallenberg Data Driven Life Science Program fellowship grant (KAW 2020.0239). M.J.S. acknowledges support by the Deutsche Forschungsgemeinschaft (STE 2360/2-1 embedded in FOR 2332) as well as ERC grant 741413 Humans on Planet Earth (HOPE). A.A. acknowledges financial support from the Swedish Research Council (2019-05191), the Swedish Foundation for Strategic Environmental Research MISTRA (Project BioPath), and the Kew Foundation. S.F. (and R.C. via S.F.) received support from the Swedish Research Council (2017-03862) and Carl Tryggers Stiftelse, project number CTS 18:105.

## Author contributions

The project was conceptualized by S.F., R.C., F.S., M.J.S. and T.M.B. Data were prepared by R.C. and F.S., and the data were analysed by R.C and T.A. The first version of the manuscript was written by R.C. and was constructively reviewed and edited by R.C., F.S., T.A., T.M.B., M.J.S., A.A. and S.F.

## Competing interests

The authors declare no competing interests.
