## [Peer Review File · Nature Communications]

Undiscovered bird extinctions obscure the true magnitude of human-driven extinction wavesReviewers' Comments:

Reviewer #1:

Remarks to the Author:

I thought this was a nicely written paper that deals with a large series of datasets (islands, radiocarbon dates, and avifauna) to postulate the number of bird extinctions on a global scale which primarily focuses on islands (for which they provide good reasons). They also develop a model for estimating what the "undiscovered" number of bird extinctions might be through statistical analysis using a well-constructed dataset as an analogue (i.e., New Zealand).

Overall, I had little to quibble with as the authors seemed to have built a robust argument based on the enormous amount of data they have compiled and the interpretations seemed valid to me. I should note, however, that I am not familiar with all of the statistical analyses they have done and so cannot comment on the paper in terms of their efficacy.

With that said, I do have some comments, which are mostly minor, that I think the authors should address prior to it being acceptable for publication.

Lines 95-96 – instead of saying "arrived", it would be better to say "translocated with them"

Line 97 – should this be "naïve"?

Lines 97-98 – "extinctions" used twice in sentence – perhaps try to use some alternative phrasing to avoid duplication

Line 99 – should be "the Marquesas Islands"

Line 100 – instead of singling out one species of moa, I would probably note for readers given the prominence that New Zealand plays in their analysis, that there are nine species that are currently recognized (I believe there used to be more based on skeletal morphology, but genetics has pared this down). In terms of mentioning New Zealand, it is becoming more common (and more appropriate) to refer to this as "Aotearoa New Zealand" in deference to the Maori.

Line 114 – should use symbol "x" instead of lowercase "x"; check throughout

Line 160 – while the authors cite Steadman's (2006) book here, I would probably add it to the list here as well (i.e., reference 11).

Line 172 – the authors mention the extinct Haast's Eagle on ANZ; it might be worth mentioning that the extinction of this species, which was the largest eagle known to have ever existed, was the likely result of trophic cascades working in tandem – a top down (humans killing moa) and bottom up (moa populations declining as the primary food source, moa, were driving to extinction).

Reference 104 – authors use only this citation in reference to early colonization of the Americas; however, the Cooper's Ferry site reported by Davis et al. (2019) is probably the most robust early site yet documented and should be referenced here as well.

Davis, L.G., Madsen, D.B., Becerra-Valdivia, L., Higham, T., Sisson, D.A., Skinner, S.M., Stueber, D., Nyers, A.J., Keen-Zebert, A., Neudorf, C. and Cheyney, M., 2019. Late upper paleolithic occupation at Cooper's Ferry, Idaho, USA, ~ 16,000 years ago. *Science*, 365(6456), pp.891-897.

Line 553 – The authors talk a lot about bird extinctions, but don't really provide a comprehensive list in one place of why they occur. I think it would be useful for readers to know that these likely occurred for a multitude of reasons, including: 1) the introduction of non-native fauna such as pigs, dogs, rats,

and chickens—at least in the case of Remote Oceania (with rats likely being the biggest culprit); landscape modification and deforestation for developing arable land for agriculture (which was crucial for successful island colonization in most, but not all instances); direct predation from humans (e.g., the dodo historically and moa in ANZ and Samoa’s star mounds for pigeon hunting are just a few prime examples during the Late Holocene); isolation that limited or prevented new individuals from replenishing depleted populations, etc.

Line 654 – While it is true that “isolation is a major driver of island biodiversity”, it might be worth mentioning that this can be easily observed in the Indo-Pacific and referred to as faunal attenuation.

Lines 658-660 – perhaps mention that Hawaii is a case par excellence?

Something missing from the list of variables in the “Data Preparation” section is soil chemistry. Under no. 5 (temperature), they note that climate can affect “fossil preservation”, but this can be mitigated by soils found in carbonate environments (e.g., atolls, makatea islands) which are much more conducive to preserving organic remains. In addition, artificially constructed environments in the form of shell middens, even when found on islands or geological substrates that are exclusively acidic, volcanic soils, can actually aid in preservation.

The following statement that “well-preserved avian fossils are more common in warm and humid climates, compared to cool or dry climates”, is more a function of diversity (i.e., the tropics have higher rates of biodiversity so it would be expected that more species would be found both extant and extinct) than it is taphonomically (i.e., cool, dry climates are more likely to preserve organic remains because tropical climates are generally not conducive to this apart from the reasons mentioned above).

Line 696 – by definition an archipelago is more than a single island.

Line 697 – The presence of native rodents is relatively few though I would think, no? That’s certainly the case for most of Remote Oceania.

Line 702 – The authors might also be interested in the more recent paper here in addition to the Boivin et al. (2019) book by Louys et al. (2021).

Louys, J., Braje, T.J., Chang, C.H., Cosgrove, R., Fitzpatrick, S.M., Fujita, M., Hawkins, S., Ingicco, T., Kawamura, A., MacPhee, R.D. and McDowell, M.C., 2021. No evidence for widespread island extinctions after Pleistocene hominin arrival. *Proceedings of the National Academy of Sciences*, 118(20), p.e2023005118.

I would probably note here too that while the authors provide a few examples of human pressures, that it’s primarily related to agriculture and land modification (both related to and unrelated to food production, such as establishing villages, etc.). It is no real surprise that most islands in prehistory are not colonized (or settled permanently) by humans until they can bring with them domesticates, especially plants that ensures a consistent source of food.

Something that I also found myself wanting to examine is the list of islands in a table along with the dates of human colonization that were used. This would also provide transparency to their dataset and allow others to conduct their own assessment. While the authors note in Extended Data Table 1 that dates for “Human Arrival” were extrapolated from references 83, 116-162, this would be an enormous task for anyone wanting to check their work. Though I am not familiar with all of the references listed here, I would like to note the following:

Reference 121 – this looks to be the sole reference for the Palauan archipelago that is included in their list of 69 focal archipelagos. This paper, however, is loaded with controversy and does not even

represent the earliest dates for these islands (though this would not affect this paper's current analysis). I recall that this paper initially received a lot of news at the time given that Berger et al. were purporting it to be a similar case of endemic dwarfism as was being made for *Homo floresiensis*. There were two subsequent rejoinders to this by Fitzpatrick et al. (2008) and Stone et al. (2019) confirming this as an erroneous interpretation. A more suitable paper to cite for the earliest Palau dates would probably be Fitzpatrick and Jew (2018) and/or Clark (2005).

Clark, G.R., 2005. A 3000-year culture sequence from Palau, western Micronesia. *Asian Perspectives*, pp.349-380.

Fitzpatrick, S.M. and Jew, N.P., 2018. Radiocarbon dating and Bayesian modelling of one of Remote Oceania's oldest cemeteries at Chelechol ra Orrak, Palau. *antiquity*, 92(361), pp.149-164.

Fitzpatrick, S.M., Nelson, G.C. and Clark, G., 2008. Small scattered fragments do not a dwarf make: Biological and archaeological data indicate that prehistoric inhabitants of Palau were normal sized. *PLoS One*, 3(8), p.e3015.

Stone, J.H., Fitzpatrick, S.M. and Napolitano, M.F., 2017. Disproving claims for small-bodied humans in the Palauan archipelago. *antiquity*, 91(360), pp.1546-1560.

As regards dates for the Caribbean for which I think there are only 4 islands/archipelagos used in the analysis (Cuba, the Bahamas, the Leeward Antilles, and Hispaniola; a "Windward" was listed too, though it did not include "Antilles" so I assumed it was something different). However, I could only find a single citation between 116-162 as referenced in Extended Data Table 1 for any Caribbean island (i.e., 122). Where did the authors obtain the dates of colonization for these other island groups? The most recent summary of radiocarbon dates for the region is by Napolitano et al. (2019) which the authors may find useful. I did notice that Puerto Rico was not included in their overall analysis, but there is also a recent review for this as well, which provides a more expanded 14C dataset.

Napolitano, M.F., DiNapoli, R.J., Stone, J.H., Levin, M.J., Jew, N.P., Lane, B.G., O'Connor, J.T. and Fitzpatrick, S.M., 2019. Reevaluating human colonization of the Caribbean using chronometric hygiene and Bayesian modeling. *Science Advances*, 5(12), p.eaar7806.

Rodríguez Ramos, R., Rodríguez López, M. and Pestle, W.J., 2023. Revision of the cultural chronology of precolonial Puerto Rico: A Bayesian approach. *PLoS One*, 18(2), p.e0282052.

Reviewer #2:

Remarks to the Author:

Cooke et al., Bird extinctions.

This is an important and interesting paper. It assesses the impact of humans on bird extinctions as our species spread across the globe, particular oceanic islands.

The problem is the question has been asked — and answered — already in two previous papers. Those papers show that the predicted extinctions concentrated in the Pacific and so are attributed to the Polynesian expansion. Moreover, those papers include not just fossil records but predictions of how many species are missing from that record. Those papers come up with broadly the same numbers and the same distribution of extinctions.

What in this paper is original?

Pimm, S. L., Moulton, M. P. & Justice, L. J. Bird extinctions in the central Pacific. *Philos. Trans. R. Soc. Lond. B. Biol. Sci.* 344, 27–33 (1994).

Curnutt, J. & Pimm, S. L. How many bird species in Hawaii and the Central Pacific before first contact? *Stud. Avian Biol.* 22, 15–30 (2001).

The two papers use two different methods. The first uses an extension of the mark-recapture method. If all the species that survive to the present day in (say) Hawai'i were found in the fossil record, then one would assume that record is complete. In fact, about half are, suggesting that there is an equal number of unknown species in addition to those species known from the fossil record.

The second uses faunal reconstruction. Almost every island above a minimum size from Hawai'i to Henderson has a unique species of rail. Larger islands have several. Pigeons and parrots didn't get to Hawai'i, but they did get to a lot of islands. So, one adds up the predicted numbers of missing pigeons, parrots, and rails.

Now, the authors may not like these numbers. They may feel — with some justification — that their methods are better. What they cannot do is to dismiss these papers with entirely superficial quotations. They must recognise previous work, compare and contrast it to their own, discuss what they think it gets right or otherwise, and how their numbers compare.

Reviewer #3:

Remarks to the Author:

Impression:

This study presents a new estimate of the total number of anthropogenic bird extinctions that includes undiscovered extinct species and calculates extinction rates through time. This is an exceptionally thorough investigation with solid argumentation, data and methodology supporting their conclusions. All underlying assumptions are explained, and the work is reproducible and of high quality. I expect this to be an important contribution to the field.

Discussion point:

The most surprising about this study is to me the lower total estimate of bird extinctions than previous research suggested. Based on those estimates I expected several thousand anthropogenic bird extinctions in total. I would recommend elaborating a bit more on this besides what is mentioned in lines 156-158. I specifically refer to the estimates of at least a thousand undiscovered extinct birds in the Pacific alone by Duncan et al. 2013 (PNAS) and the estimate of at least 2000 extinct rails in the Pacific islands by Steadman 1995 (Science). Can you compare their arguments for arriving at those higher estimates with your arguments and methods?

About the title:

This is a bit confusing because it appears to refer to a comparison with extinctions in other species groups, which this study does not do.

Minor comment:

These sentences appear to be incomplete: Lines 174-175 and 590.

Response-to-Reviewers

Dear Reviewer 1, Reviewer 2 and Reviewer 3

Thank you very much for your reviews of our manuscript and for the constructive feedback. We agree with the revisions recommended and have made changes in response.

Overall, we have:

- Expanded our introduction of potential extinction mechanisms for birds
- Provided more detail on the novelty of our work and how it compares and contrasts to previous work throughout the manuscript
- Addressed minor comments

Specific changes are detailed in blue text below each corresponding point of feedback.

Thank you for these valuable recommendations which have helped to improve our manuscript.

Sincerely,

Rob Cooke (on behalf of all authors)

Reviewer #1 (Remarks to the Author):

I thought this was a nicely written paper that deals with a large series of datasets (islands, radiocarbon dates, and avifauna) to postulate the number of bird extinctions on a global scale which primarily focuses on islands (for which they provide good reasons). They also develop a model for estimating what the “undiscovered” number of bird extinctions might be through statistical analysis using a well-constructed dataset as an analogue (i.e., New Zealand).

Overall, I had little to quibble with as the authors seemed to have built a robust argument based on the enormous amount of data they have compiled and the interpretations seemed valid to me. I should note, however, that I am not familiar with all of the statistical analyses they have done and so cannot comment on the paper in terms of their efficacy.

Thank you for your positive comments! Always nice to hear!

With that said, I do have some comments, which are mostly minor, that I think the authors should address prior to it being acceptable for publication.

We have addressed all your thoughtful comments below.

Lines 95-96 – instead of saying “arrived”, it would be better to say “translocated with them”

Good point! We have made the suggested change.

Line 97 – should this be “naïve”?

We have changed naive to naïve throughout the manuscript for consistency with the literature.

Lines 97-98 – “extinctions” used twice in sentence – perhaps try to use some alternative phrasing to avoid duplication

We have deleted one of the extinctions from this sentence to avoid duplication: **“Known extinctions potentially associated with this wave include ...”**

Line 99 – should be “the Marquesas Islands”

Thanks, we have made this change.

Line 100 – instead of singling out one species of moa, I would probably note for readers given the prominence that New Zealand plays in their analysis, that there are nine species that are currently recognized (I believe there used to be more based on skeletal morphology, but genetics has pared this down). In terms of mentioning New Zealand, it is becoming more common (and more appropriate) to refer to this as “Aotearoa New Zealand” in deference to the Maori.

Good points! We have changed the text to:

“Known extinctions potentially associated with this wave include the High-billed Crow (*Corvus imfluviatus*) from Hawaii, Sinoto’s Lorikeet (*Vini sinotoi*) from the Marquesas Islands, and nine Moa (*Dinornithiformes*) species from Aotearoa New Zealand (Fig. 3c).”

We have also changed New Zealand to Aotearoa New Zealand throughout the manuscript.

Line 114 – should use symbol “x” instead of lowercase “x”; check throughout

We have changed x to × throughout the manuscript.

Line 160 – while the authors cite Steadman’s (2006) book here, I would probably add it to the list here as well (i.e., reference 11).

We have now added the Steadman reference to this sentence.

Line 172 – the authors mention the extinct Haast’s Eagle on ANZ; it might be worth mentioning that the extinction of this species, which was the largest eagle known to have ever existed, was the likely result of trophic cascades working in tandem – a top down (humans killing moa) and bottom up (moa populations declining as the primary food source, moa, were driving to extinction).

In this section we are highlighting some of the key ecosystem functions contributed by extinct bird species and felt that there wasn’t space to explain the extinction process of the Haast’s Eagle here, instead we have directed the reader to ref. ³⁷ (Holdaway, R.N., 1989. New Zealand’s pre-human avifauna and its vulnerability. *New Zealand journal of ecology*, pp.11-25.) which provides a more detailed explanation of their decline. We have also expanded our information on drivers of bird extinctions in the introduction (see response to comment below), now covering multiple mechanisms.

Reference 104 – authors use only this citation in reference to early colonization of the Americas; however, the Cooper’s Ferry site reported by Davis et al. (2019) is probably the most robust early site yet documented and should be referenced here as well.

Davis, L.G., Madsen, D.B., Becerra-Valdivia, L., Higham, T., Sisson, D.A., Skinner, S.M., Stueber, D., Nyers, A.J., Keen-Zebert, A., Neudorf, C. and Cheyney, M., 2019. Late upper paleolithic occupation at Cooper’s Ferry, Idaho, USA, ~ 16,000 years ago. *Science*, 365(6456), pp.891-897.

Thank you, we have now added that paper as a reference.

Line 553 – The authors talk a lot about bird extinctions, but don't really provide a comprehensive list in one place of why they occur. I think it would be useful for readers to know that these likely occurred for a multitude of reasons, including: 1) the introduction of non-native fauna such as pigs, dogs, rats, and chickens—at least in the case of Remote Oceania (with rats likely being the biggest culprit); landscape modification and deforestation for developing arable land for agriculture (which was crucial for successful island colonization in most, but not all instances); direct predation from humans (e.g., the dodo historically and moa in ANZ and Samoa's star mounds for pigeon hunting are just a few prime examples during the Late Holocene); isolation that limited or prevented new individuals from replenishing depleted populations, etc.

We have now expanded on a single sentence we originally had in the Introduction to better describe the drivers of extinction for birds:

“Even small human populations rapidly devastated island avifaunas as they introduced new threats outside the evolutionary experience of native species². Drivers of human-driven bird extinctions include habitat loss associated with land clearance (cutting, burning) and the introduction of non-native plants and crops⁴, the introduction of alien species (domestic animals and/or human commensals)^{2,5,6} and the overexploitation of bird species via hunting and trapping (birds were hunted for their fat, protein, bones and colourful feathers)^{1,5}.”

We believe this is a useful addition for the reader, so thank you for this suggestion. We have also added more text on human arrival and the associated threatening processes to the Methods (see response to comment below).

Line 654 – While it is true that “isolation is a major driver of island biodiversity”, it might be worth mentioning that this can be easily observed in the Indo-Pacific and referred to as faunal attenuation.

We decided not to directly mention ‘faunal attenuation’ here as we find the term slightly unclear and not commonly used. Instead, we have reworded the sentence to hopefully make our point more clearly:

“Isolation is a major driver of island biodiversity, as isolation facilitates in-situ speciation, which leads to high levels of endemism in islands (despite islands tending to have lower species richness overall)^{31,61}.”

We also refer readers to the references that go into more detail on the relationship between isolation and island biodiversity (MacArthur, R. H. & Wilson, E. O. The theory of island biogeography. Princeton University Press, 1967; and Whittaker, R. J. & Fernández-Palacios, J. M. Island Biogeography: Ecology, Evolution, and Conservation. Oxford University Press, 2007).

Lines 658-660 – perhaps mention that Hawaii is a case par excellence?

Agreed, we think that is a helpful addition for the reader, thanks.

“Archipelagos that are more isolated are likely to have lower rates of colonization, leading to fewer species overall, but higher rates of endemism⁶⁴, and thus greater probability of global, rather than regional, extinction. A classic example of this phenomenon is Hawaii⁶⁵.”

Something missing from the list of variables in the “Data Preparation” section is soil chemistry. Under no. 5 (temperature), they note that climate can affect “fossil preservation”, but this can be mitigated by soils found in carbonate environments (e.g., atolls, makatea islands) which are much more conducive to preserving organic remains. In addition, artificially constructed environments in the

form of shell middens, even when found on islands or geological substrates that are exclusively acidic, volcanic soils, can actually aid in preservation.

We did consider soil properties as possible variables to include in our models, however available data were unsuitable. We have now clarified this in the manuscript for readers:

“Although soil chemistry is known to affect fossil preservation^{54,55}, suitable data at the global scale are too imprecise and the influence of soil properties on preservation is often local and complex⁵⁵.”

The following statement that “well-preserved avian fossils are more common in warm and humid climates, compared to cool or dry climates”, is more a function of diversity (i.e., the tropics have higher rates of biodiversity so it would be expected that more species would be found both extant and extinct) than it is taphonomically (i.e., cool, dry climates are more likely to preserve organic remains because tropical climates are generally not conducive to this apart from the reasons mentioned above).

We fully agree that this is a more complex pattern than the text initially suggested. We based this statement directly on the reference cited (Gardner, E.E., Walker, S.E. and Gardner, L.I., 2016. Palaeoclimate, environmental factors, and bird body size: a multivariable analysis of avian fossil preservation. *Earth-Science Reviews*, 162, pp.177-197). “The climate zone data reveal significantly greater skeletal completeness for specimens in warm and humid climates when compared to cool or dry climates.” However, the authors also state “the mechanisms that explain the warm climate effect remain unclear”, they subsequently hypothesize the effect of freeze/thaw, or the effect of vegetation covering the carcass. Since the effect of climate is complex and likely indirect we have deleted this sentence.

Line 696 – by definition an archipelago is more than a single island.

We have changed this to **“In addition, SD area captures differences in structure between archipelagos, e.g., archipelagos with a few large islands vs many small islands.”**

Line 697 – The presence of native rodents is relatively few though I would think, no? That’s certainly the case for most of Remote Oceania.

It’s a good point that most of remote Oceania does not have native rodents. Still, over a third of the focal archipelagos (24 of 68) have native rodents and as this is such a clear potential mechanism of extinction for birds, we think it is important to include. It is also worth noting that it is not a conceptual problem for our analyses if we include predictors that turn out to be unimportant. We have added this detail to the manuscript:

“The impact of (human-introduced) exotic predators, particularly rats and mice, is reduced on islands that possess native rodents, likely because these avifaunas are less naïve to the effects of nest predation^{72,73}. Thus, insufficient eco-evolutionary exposure (i.e., absence of native rodents) likely drives avian naïvety towards exotic predators⁷², and therefore relates to extinction risk. Approximately a third of the focal archipelagos have native rodents and hence likely reduced extinction risk from exotic predators.”

Line 702 – The authors might also be interested in the more recent paper here in addition to the Boivin et al. (2019) book by Louys et al. (2021).

Louys, J., Braje, T.J., Chang, C.H., Cosgrove, R., Fitzpatrick, S.M., Fujita, M., Hawkins, S., Ingicco, T., Kawamura, A., MacPhee, R.D. and McDowell, M.C., 2021. No evidence for widespread island

extinctions after Pleistocene hominin arrival. *Proceedings of the National Academy of Sciences*, 118(20), p.e2023005118.

Thanks for bringing our attention to this paper. We have now added it as a reference (see next response – ref. ⁷⁶).

I would probably note here too that while the authors provide a few examples of human pressures, that it's primarily related to agriculture and land modification (both related to and unrelated to food production, such as establishing villages, etc.). It is no real surprise that most islands in prehistory are not colonized (or settled permanently) by humans until they can bring with them domesticates, especially plants that ensures a consistent source of food.

We have now expanded the text to more clearly relate differences in first human arrival to potential changes in human pressures:

“In addition, the timing of peopling of an archipelago could affect the magnitude and structure of human pressures^{6,75,76}. Indeed, human arrival could reflect changes in agricultural practice and land modification (widespread land clearance and domesticated plants are more often associated with later human settlements)⁷⁵, changes in hunting preferences, technologies and efficiency (e.g., use of bows, arrows, and spears; fishing capabilities)^{76,77}, changes in the number of human introductions and commensals (e.g., pigs, rats)^{2,5,6}, and changes in human behaviour and culture (e.g., communication/language, long-distance trade)^{75,76}.”

Something that I also found myself wanting to examine is the list of islands in a table along with the dates of human colonization that were used. This would also provide transparency to their dataset and allow others to conduct their own assessment. While the authors note in Extended Data Table 1 that dates for “Human Arrival” were extrapolated from references 83, 116-162, this would be an enormous task for anyone wanting to check their work.

That's a good point. We provided a csv of human first arrival dates and their references on our GitHub repository (https://github.com/03rcooke/ext_birds/blob/v1.0.0/data/colz.csv), but it was not clear where to find this, so apologies! We have now added the information as an additional Supplementary Table (see Supplementary Table 1 in the Supplementary Information; too large to include here). As well as providing it within the code repository; colz.csv at <https://zenodo.org/record/8270298>.

“We compiled the date of first human arrival from the literature (Supplementary Table 1).”

We have also added a table of the islands within each archipelago to the code repository. Making it clear where to find this as well:

“We grouped these 1,488 islands into 69 archipelagos (Supplementary Fig. 2) according to an existing classification⁴³, with minor modifications (see isl_arch.csv at <https://zenodo.org/record/8270298> for details on islands within archipelagos).”

Though I am not familiar with all of the references listed here, I would like to note the following:

Reference 121 – this looks to be the sole reference for the Palauan archipelago that is included in their list of 69 focal archipelagos. This paper, however, is loaded with controversy and does not even represent the earliest dates for these islands (though this would not affect this paper's current analysis). I recall that this paper initially received a lot of news at the time given that Berger et al. were purporting it to be a similar case of endemic dwarfism as was being made for *Homo floresiensis*. There were two subsequent rejoinders to this by Fitzpatrick et al. (2008) and Stone et al.

(2019) confirming this as an erroneous interpretation. A more suitable paper to cite for the earliest Palau dates would probably be Fitzpatrick and Jew (2018) and/or Clark (2005).

Clark, G.R., 2005. A 3000-year culture sequence from Palau, western Micronesia. *Asian Perspectives*, pp.349-380.

Fitzpatrick, S.M. and Jew, N.P., 2018. Radiocarbon dating and Bayesian modelling of one of Remote Oceania's oldest cemeteries at Chelechol ra Orrak, Palau. *antiquity*, 92(361), pp.149-164.

Fitzpatrick, S.M., Nelson, G.C. and Clark, G., 2008. Small scattered fragments do not a dwarf make: Biological and archaeological data indicate that prehistoric inhabitants of Palau were normal sized. *PLoS One*, 3(8), p.e3015.

Stone, J.H., Fitzpatrick, S.M. and Napolitano, M.F., 2017. Disproving claims for small-bodied humans in the Palauan archipelago. *antiquity*, 91(360), pp.1546-1560.

Thank you for your expertise on this matter. As these papers came with similar dates of first human arrival, we originally decided to just use one as the reference (the Berger et al. paper), but we were unaware of the controversy surrounding this paper. We have now replaced the reference with Fitzpatrick and Jew 2018, which uses more modern methods but comes to a similar conclusion on the date of first human arrival.

As regards dates for the Caribbean for which I think there are only 4 islands/archipelagos used in the analysis (Cuba, the Bahamas, the Leeward Antilles, and Hispaniola; a “Windward” was listed too, though it did not include “Antilles” so I assumed it was something different). However, I could only find a single citation between 116-162 as referenced in Extended Data Table 1 for any Caribbean island (i.e., 122). Where did the authors obtain the dates of colonization for these other island groups? The most recent summary of radiocarbon dates for the region is by Napolitano et al. (2019) which the authors may find useful. I did notice that Puerto Rico was not included in their overall analysis, but there is also a recent review for this as well, which provides a more expanded 14C dataset.

Napolitano, M.F., DiNapoli, R.J., Stone, J.H., Levin, M.J., Jew, N.P., Lane, B.G., O’Connor, J.T. and Fitzpatrick, S.M., 2019. Reevaluating human colonization of the Caribbean using chronometric hygiene and Bayesian modeling. *Science Advances*, 5(12), p.eaar7806.

Rodríguez Ramos, R., Rodríguez López, M. and Pestle, W.J., 2023. Revision of the cultural chronology of precolonial Puerto Rico: A Bayesian approach. *PLoS One*, 18(2), p.e0282052.

We included five Caribbean archipelagos in our analysis (Bahamas* = Bahamas, and Turks and Caicos; Cuba* = Cuba, Cayman Islands, and Jamaica; Hispaniola* = Hispaniola, Puerto Rico, U.S. Virgin Islands and, British Virgin Islands; Leeward Antilles = Leeward Antilles; and Windward* = Windward and Leeward Islands; these definitions are available in Supplementary Table 5 and are mapped as Supplementary Fig. 2 – there is not enough space in the text to explain these more fully unfortunately). Puerto Rico was therefore included in the Hispaniola* archipelago, but the name refers to the largest island in the archipelago. For these archipelagos we used a previously published compilation of first human arrival dates (Kouvari, M. and van der Geer, A.A., 2018. Biogeography of extinction: The demise of insular mammals from the Late Pleistocene till today. *Palaeogeography, Palaeoclimatology, Palaeoecology*, 505, pp.295-304.), which had dates in its Supplementary Materials for many of the Caribbean islands. It did not have dates for any of the islands in the Bahamas* archipelago. So, for this archipelago we used a separate reference (Berman, M.J. and Gnivecki, P.L.,

1995. The colonization of the Bahama archipelago: A reappraisal. *World Archaeology*, 26(3), pp.421-441) as noted by you.

We compiled data on first human arrival before Napolitano was published, which is a shame, however there is strong agreement between Kouvari and Napolitano, and these differences are unlikely to impact the analyses. Specifically, Kouvari estimates first arrival as 6000-4000 years BP for all Caribbean archipelagos, whereas Napolitano estimates ~5400-4700 years BP for Cuba*, ~4600-3900 for Hispaniola*, ~5700-4800 for Leeward Antilles and ~5900-4400 for Windward*. The less precise estimates for Kouvari cover the more precise Napolitano estimates and usefully capture some of the within-archipelago variation in first human arrival and we therefore decided to remain with the coarser Kouvari estimate.

Thanks again for your constructive and helpful comments!

Reviewer #2 (Remarks to the Author):

Cooke et al., Bird extinctions.

This is an important and interesting paper. It assesses the impact of humans on bird extinctions as our species spread across the globe, particular oceanic islands.

Thank you! We agree that this is an important topic.

The problem is the question has been asked — and answered — already in two previous papers. Those papers show that the predicted extinctions concentrated in the Pacific and so are attributed to the Polynesian expansion. Moreover, those papers include not just fossil records but predictions of how many species are missing from that record. Those papers come up with broadly the same numbers and the same distribution of extinctions.

What in this paper is original?

Pimm, S. L., Moulton, M. P. & Justice, L. J. Bird extinctions in the central Pacific. *Philos. Trans. R. Soc. Lond. B. Biol. Sci.* 344, 27–33 (1994).

Curnutt, J. & Pimm, S. L. How many bird species in Hawaii and the Central Pacific before first contact? *Stud. Avian Biol.* 22, 15–30 (2001).

The two papers use two different methods. The first uses an extension of the mark-recapture method. If all the species that survive to the present day in (say) Hawai'i were found in the fossil record, then one would assume that record is complete. In fact, about half are, suggesting that there is an equal number of unknown species in addition to those species known from the fossil record.

The second uses faunal reconstruction. Almost every island above a minimum size from Hawai'i to Henderson has a unique species of rail. Larger islands have several. Pigeons and parrots didn't get to Hawai'i, but they did get to a lot of islands. So, one adds up the predicted numbers of missing pigeons, parrots, and rails.

Now, the authors may not like these numbers. They may feel — with some justification — that their methods are better. What they cannot do is to dismiss these papers with entirely superficial quotations. They must recognise previous work, compare and contrast it to their own, discuss what they think it gets right or otherwise, and how their numbers compare.

We agree that these are key papers! We also agree that they ask similar questions to the questions we tackle, along with two other key papers (Steadman 1995 and Duncan et al., 2013). We have now

attempted to highlight this important body of previous work more clearly and how we have built upon it.

Specifically, we have flagged these papers explicitly for the reader in the Introduction:

“Hence, the full extent of bird extinctions since the Late Pleistocene remains unknown (although see previous studies of the Pacific^{4,7,10,12}).”

Have outlined our advances:

“Here, we go beyond previous studies of the Pacific^{4,7,10,12} to cover the globe, evaluating the spatial distribution of extinctions, and incorporating undiscovered extinctions, and their uncertainty, into temporal analyses of extinction rate. Hence, we quantify the total magnitude, distribution and rate of bird extinctions worldwide since the Late Pleistocene, including undiscovered extinctions.”

We have also explicitly added the context of the Pacific vs global estimates to our Abstract and Results.

Abstract: **“We estimate that the Pacific accounts for 61% of total bird extinctions.”**

Results: **“For the Pacific we estimate 875 (773 – 973) total bird extinctions with 554 (450 – 652) undiscovered, compared to 557 (508 – 605) total extinctions outside the Pacific with 235 (185 – 281) undiscovered. Hence, we estimate that the Pacific accounts for 61% of total bird extinctions and 70% of undiscovered bird extinctions.”**

Finally, although we attempted to compare and contrast our results to these previous estimates in our original Discussion, we agree that this was not sufficient to really explore the similarities and differences. So, to flesh this out better we have expanded our paragraph in the Discussion, which now reads:

“Previous research has estimated the number of bird extinctions across the Pacific^{4,7,10,12}, with the expectation that most extinctions would be located there (confirmed here as 61% of total bird extinctions). Specifically, these studies use faunal reconstruction^{4,10} and variants of mark-recapture^{7,12} to estimate bird extinctions. These estimates range from ~800¹², to ~1,300⁷, to over 2,000⁴ total Pacific bird extinctions, although definitions of the Pacific vary (i.e., which islands are included/excluded), and which bird groups were studied also differ, making direct comparisons difficult. Still, for the Pacific we estimate 875 (95% Credible Interval: 773 – 973) total bird extinctions, which falls within the lower to mid-range of previous estimates^{4,7,10,12}. Our Pacific estimate is similar to a previous study that suggested there were more than ~600 but fewer than ~1300 undiscovered extinctions¹⁰. By contrast, our estimate is lower than a recent estimate of ~1,300 for all birds⁷, although this study also presents a median estimate for only non-passerine landbirds of 983 (95% Credible Interval: 731 – 1,332). When considering the uncertainty of both estimates there is strong overlap in the credible intervals, highlighting their general agreement. Our estimate is similar to all previous estimates except for the estimate of over 2,000 undiscovered extinctions⁴. This number has been previously described as a likely overestimate^{10,30}, due to the occurrence of natural events (e.g., volcanism, tsunamis) across some islands assumed to host undiscovered endemic birds that might have prevented colonization or extirpated populations before speciation occurred^{10,30,31}. In addition, our estimate of the ratio of undiscovered to discovered extinctions for the Pacific of 0.63 (95% Credible Interval: 0.59 – 0.67) is similar to previous estimates of 0.5¹² and 0.67 (95% Credible Interval: 0.46 – 0.84)⁷. Thus, our estimate is supported by a range of datasets and analytical techniques, both within our study (see Methods) and across previous studies^{4,7,10,12}.”

Reviewer #3 (Remarks to the Author):

Impression:

This study presents a new estimate of the total number of anthropogenic bird extinctions that includes undiscovered extinct species and calculates extinction rates through time. This is an exceptionally thorough investigation with solid argumentation, data and methodology supporting their conclusions. All underlying assumptions are explained, and the work is reproducible and of high quality. I expect this to be an important contribution to the field.

Thank you very much, we did attempt to be as thorough as possible!

Discussion point:

The most surprising about this study is to me the lower total estimate of bird extinctions than previous research suggested. Based on those estimates I expected several thousand anthropogenic bird extinctions in total. I would recommend elaborating a bit more on this besides what is mentioned in lines 156-158. I specifically refer to the estimates of at least a thousand undiscovered extinct birds in the Pacific alone by Duncan et al. 2013 (PNAS) and the estimate of at least 2000 extinct rails in the Pacific islands by Steadman 1995 (Science). Can you compare their arguments for arriving at those higher estimates with your arguments and methods?

Good point! We have now greatly expanded on the section in the Discussion describing previous estimates and how they compare to our estimate:

“Previous research has estimated the number of bird extinctions across the Pacific^{4,7,10,12}, with the expectation that most extinctions would be located there (confirmed here as 61% of total bird extinctions). Specifically, these studies use faunal reconstruction^{4,10} and variants of mark-recapture^{7,12} to estimate bird extinctions. These estimates range from ~800¹², to ~1,300⁷, to over 2,000⁴ total Pacific bird extinctions, although definitions of the Pacific vary (i.e., which islands are included/excluded), and which bird groups were studied also differ, making direct comparisons difficult. Still, for the Pacific we estimate 875 (95% Credible Interval: 773 – 973) total bird extinctions, which falls within the lower to mid-range of previous estimates^{4,7,10,12}. Our Pacific estimate is similar to a previous study that suggested there were more than ~600 but fewer than ~1300 undiscovered extinctions¹⁰. By contrast, our estimate is lower than a recent estimate of ~1,300 for all birds⁷, although this study also presents a median estimate for only non-passerine landbirds of 983 (95% Credible Interval: 731 – 1,332). When considering the uncertainty of both estimates there is strong overlap in the credible intervals, highlighting their general agreement. Our estimate is similar to all previous estimates except for the estimate of over 2,000 undiscovered extinctions⁴. This number has been previously described as a likely overestimate^{10,30}, due to the occurrence of natural events (e.g., volcanism, tsunamis) across some islands assumed to host undiscovered endemic birds that might have prevented colonization or extirpated populations before speciation occurred^{10,30,31}. In addition, our estimate of the ratio of undiscovered to discovered extinctions for the Pacific of 0.63 (95% Credible Interval: 0.59 – 0.67) is similar to previous estimates of 0.5¹² and 0.67 (95% Credible Interval: 0.46 – 0.84)⁷. Thus, our estimate is supported by a range of datasets and analytical techniques, both within our study (see Methods) and across previous studies^{4,7,10,12}.”

About the title:

This is a bit confusing because it appears to refer to a comparison with extinctions in other species groups, which this study does not do.

That's true, in response we have changed the title to **"Undiscovered bird extinctions obscure the true magnitude of human-driven extinction waves"**, removing the comparison to other taxonomic groups.

Minor comment:

These sentences appear to be incomplete: Lines 174-175 and 590.

Apologies for this, we have now completed both sentences:

"We estimate ~800 undiscovered bird extinctions, many of which could also have had key roles in their ecosystems."

"We compiled previous estimates of island area and connection to the mainland⁴⁴, where connection to the mainland was based on global bathymetry data⁴⁶ with a sea level of -122m during the Last Glacial Maximum^{44,45}."

Reviewers' Comments:

Reviewer #1:

Remarks to the Author:

I have read through the revised manuscript and the authors' responses to all of my original comments. I think they have done a very nice job at responding to these and believe that the paper is publishable in its current form, pending satisfactory responses to comments from the other reviewer.

Reviewer #2:

Remarks to the Author:

The authors now acknowledge that this is far from an original contribution. It builds on extensive work done by others on the Pacific extinctions that found very similar results. (The Pacific is where more of the extinctions occur.). The authors consider a larger geographical extent, which is novel, and they add details to the overall story. In that sense, it adds to the existing literature.

Reviewer #3:

Remarks to the Author:

My concerns have been addressed sufficiently in the revision.